# Sparse activity of identified dentate granule cells during spatial exploration

Maria Diamantaki[1,2], Markus Frey[1], Philipp Berens[1,3,4], Patricia Preston-Ferrer[1]*, Andrea Burgalossi[1]*

[1]Werner-Reichardt Centre for Integrative Neuroscience, University of Tübingen, Tübingen, Germany; [2]Graduate Training Centre of Neuroscience - IMPRS, University of Tübingen, Tübingen, Germany; [3]Institute for Ophthalmic Research, University of Tübingen, Tübingen, Germany; [4]Bernstein Centre for Computational Neuroscience, University of Tübingen, Tübingen, Germany

**Abstract** In the dentate gyrus – a key component of spatial memory circuits – granule cells (GCs) are known to be morphologically diverse and to display heterogeneous activity profiles during behavior. To resolve structure–function relationships, we juxtacellularly recorded and labeled single GCs in freely moving rats. We found that the vast majority of neurons were silent during exploration. Most active GCs displayed a characteristic spike waveform, fired at low rates and showed spatial activity. Primary dendritic parameters were sufficient for classifying neurons as active or silent with high accuracy. Our data thus support a sparse coding scheme in the dentate gyrus and provide a possible link between structural and functional heterogeneity among the GC population.

*For correspondence: patricia. preston@cin.uni-tuebingen.de (PP-F); andrea.burgalossi@cin.uni-tuebingen.de (AB)

Competing interests: The authors declare that no competing interests exist.

## Introduction

Sparse activity is a hallmark feature of cortical and hippocampal function (*Treves and Rolls, 1994*; *Olshausen and Field, 2004*; *Wolfe et al., 2010*; *Barth and Poulet, 2012*; *Spanne and Jörntell, 2015*). In neocortical circuits, sparse representations are thought to be crucial for optimal encoding of sensory stimuli (*Brecht et al., 2004*; *Huber et al., 2008*; *Li et al., 2009*; *Rochefort et al., 2009*). Similarly in hippocampal circuits, the skewed distribution of population firing activity indicates that a preconfigured, highly active minority of neurons dominates information coding (*Mizuseki and Buzsáki, 2013*; *Buzsaki and Mizuseki, 2014*). Whether and how this highly active population is structurally or molecularly different from the majority of neurons remains to be established. The dentate gyrus (DG) is among the most sparse networks known to date. Extracellular recordings in foraging rodents have in fact revealed that only a tiny minority of principal neurons – putative granule cells (GCs) – is active during behavior, while the rest are silent (*Jung and McNaughton, 1993*; *Skaggs et al., 1996*; *Neunuebel and Knierim, 2012*). Although several structure–function schemes have been proposed (*Leutgeb et al., 2007*; *Alme et al., 2010*; *Lisman, 2011*; *Neunuebel and Knierim, 2014*), the cellular identity of active and silent neurons has remained largely unexplored.

Here, we addressed this issue by employing juxtacellular recording and labeling procedures in freely moving rats (*Tang et al., 2014a*). An advantage of this technique over conventional extracellular methods is that neurons can be identified irrespective of their spiking activity; thus, both silent and active cells can be sampled during free behavior (i.e. under intact sensory, motor and vestibular inputs), which enables the analysis of sparsely active networks (*Hromádka et al., 2008*; *O'Connor et al., 2010*; *Herfst et al., 2012*; *Burgalossi et al., 2014*). We provide a quantitative assessment of sparse activity in the rat's DG and show that a subset of mature GCs with complex dendritic trees contributes to the active pool.

## Results

In a first set of experiments, we aimed to explore electrophysiological signatures of GC activity. To this end, we juxtacellularly recorded and labeled spontaneously active neurons in the rat DG in-vivo. From these recordings, we observed that active GCs displayed a characteristic spike waveform, as shown in the representative example in *Figure 1A–C*. Here, a spontaneously active GC was recorded and identified in a head-fixed awake rat. This neuron displayed morphological features of mature GCs (see Materials and methods and *Figure 1A*) and discharged sparsely during the awake state (average firing rate=0.49 Hz; *Figure 1B*). Notably, spikes from this neuron displayed a 'shoulder' following the positive peak of the juxtacellular spike, which was best evident as a local maximum in the first derivative of the juxtacellular voltage trace (*Figure 1C*). Altogether, we recorded and identified 47 active neurons during urethane theta-oscillations (n=19), awake head-fixation (n=4) and free-behavior (n=24) (*Figure 1D*). In these recordings, the presence of spike-shoulders was determined by the intersection between the spike trace and its moving average during the repolarization phase of the spike (see *Figure 1—figure supplement 1*). Irrespective of the recording condition (i.e. anesthetized versus awake), we found that all neurons displaying spike-shoulders were located within the GC layer (n=17; with the exception of one excitatory neuron just below the GC layer; see *Figure 1D, E* and Materials and methods) and corresponded to morphologically mature GCs (see criteria in Materials and methods; n=15 cases where morphology could be assessed). In CA3c (n=7), fast-spiking (n=10) and hilar (n=17) neurons, spike-shoulders were not identified (*Figure 1F*). In most cases spike-shoulders from GCs were already apparent in the average spike waveform (*Figure 1C,E* and *Figure 1—figure supplement 1A*) and resulted in a relatively long duration of the juxtacellular spike signal (*Figure 1G*). Furthermore, recordings could be assigned with high accuracy to the GC class based solely on the average spike waveforms (~89%; cross-validated; see Materials and methods for details), thus providing additional evidence for a correlation between spike-shoulders and GC identity.

Next, we performed juxtacellular recordings from single DG neurons while rats explored an open-field environment or an elevated 'O' maze. Of 190 juxtacellular recordings established in the GC layer, 163 were from silent neurons (see Materials and methods and *Figure 2*). *Figure 2A–D* shows two examples of identified silent GCs, recorded during free behavior. Both neurons displayed morphological features of mature GCs (*Figure 2A,C*) and did not fire during exploration (*Figure 2B, D*). Spiking was readily induced by current injection (*Figure 2E*), which was used in a subset of cases to label the neurons at the end of the recording session (see Materials and methods; *Pinault, 1996*; *Herfst et al., 2012*; *Tang et al., 2014a*).

Active GCs were thus very sparse and accounted for only ~14% (27 out of 190) of all neurons sampled within the GC layer (*Figure 2F*; we note that this is likely to represent an overestimate, since silent juxtacellular 'hits' were often discarded; see Materials and methods). Nine spontaneously active GCs were successfully labeled and identified in freely moving animals; consistent with previous observations (*Jung and McNaughton, 1993*; *Skaggs et al., 1996*), the sharpest spatial activity was observed within the subset of neurons displaying highest firing rates (>1 Hz); two of these neurons are shown in *Figure 3A–H*. The first morphologically mature GC (*Figure 3A*) was recorded in an open square maze and displayed a single firing field (*Figure 3B*). A large fraction of spikes (*Figure 3C*) occurred within bursts (*Figure 3D*; see also *Pernía-Andrade and Jonas, 2014*) and was rhythmically entrained by local field potential (LFP) theta oscillations (4–12Hz; *Figure 3E*). The second neuron was also a mature GC (*Figure 3F*). It was recorded while the rat explored an 'O' maze, and a single firing field was observed (*Figure 3G*). Spikes of this identified neuron also displayed shoulders (*Figure 3H*).

Given the correlation between spike-shoulders and GC identity (see *Figure 1*), we sought to take advantage of this waveform signature (together with the anatomical location of the electrode track in the GC layer, see Materials and methods) for classifying unidentified juxtacellular recordings obtained in freely moving animals. An example of a classified GC recording is shown in *Figure 3I–K*. In this recording, cell identification by juxtacellular labeling failed, but the recording site could be localized to the GC layer (see *Figure 3I* and Materials and methods). This neuron fired only a few spikes during behavior, which tended to cluster at a single spatial location (*Figure 3J*). Spikes fired by this neuron displayed prominent shoulders (*Figure 3K*), a strong correlate of GC identity (*Figure 1*). Altogether, we recorded 27 active GCs (9 identified and 18 classified) during spatial

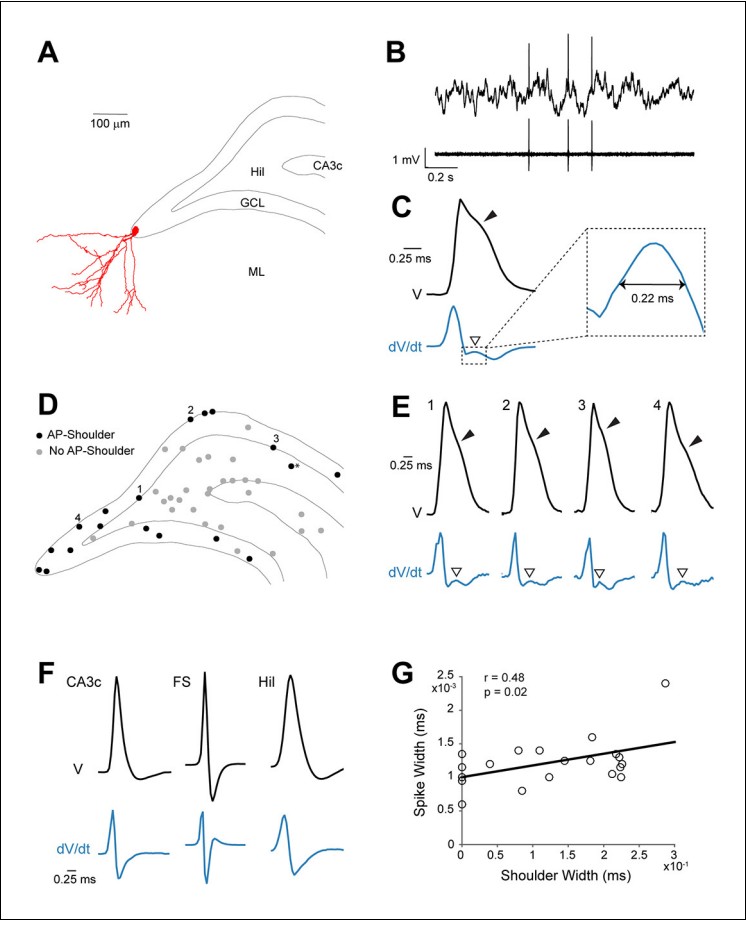

**Figure 1.** Spike waveform features of identified granule cells. (**A**) Reconstruction of the somatodendritic morphology of a morphologically mature GC recorded in an awake, head-fixed animal. (**B**) Representative raw (top) and high-pass filtered (bottom) spike trace recorded under head-fixation from the cell shown in (**A**). (**C**) Average spike waveform (black) of the cell shown in (**A**). Note the presence of the shoulder (black arrowhead) on the spike waveform and the corresponding peak (white arrowhead) in the first derivative below (blue, dV/dt). The right panel shows a magnification of the inset on the left. The 'shoulder width' measured at half-maximum of the average spike waveform's first derivative is illustrated by the double-headed arrow. The first derivative is scaled up for illustration purposes. (**D**) Schematic outline of the dentate gyrus showing the somatic location of all active neurons recorded and identified juxtacellularly (n=47 neurons: GCs, n=21; hilar cells, n=17; CA3c cells, n=7; and fast-spiking cells, n=2; see details in Materials and methods). Recordings where spike-shoulders ('AP-shoulders') were or were not detected are indicated as black or grey dots, respectively. The asterisk indicates the neuron located below the GC layer, where shoulders were also identified (see Materials and methods for details). (**E**) Representative normalized average spike waveforms (black) of four identified GCs with their corresponding first derivatives (blue). Black arrowheads indicate the spike-shoulder and white arrowheads the local peak in the first derivatives. First derivatives are scaled up for illustration purposes. The location of these cells within the GC layer is indicated by the corresponding numbers in (**D**). (**F**) Representative normalized average spike waveforms (black) of an identified CA3c cell, an identified fast-spiking cell (FS) and an identified hilar cell (Hil) with their corresponding first derivatives (blue). Note the absence of shoulders in the spike waveforms and first derivatives. First derivatives are scaled up for illustration purposes. (**G**) Correlation between spike width and 'shoulder width' for all identified GCs (n=21, see panel **D**). Regression line (black), Pearson's correlation coefficient (r) and p-value are indicated. AP=action potential; Hil=hilus; GCL=granule cell layer; ML=molecular layer.

The following figure supplement is available for figure 1:

**Figure supplement 1.** Detection of spike-shoulders.

exploration. We note that since most but not all identified active GCs displayed spike-shoulders (18 out of 21; ~86%, *Figure 1D*), a fraction of GC recordings (~14%) is likely to be misclassified as false-negative in our dataset. In line with the examples shown in *Figure 3,* in the majority of cases firing occurred at single spatial locations (8 out of 11 recordings; *Figure 3—figure supplement 1*; see details in Materials and methods), suggesting that these spatial firing patterns are likely to arise from the mature GC population (*Danielson et al., 2016*).

Next, we explored whether the heterogeneity of GC activity during behavior might relate to the morphological properties of the neurons. To this end, we first reconstructed the somato-dendritic

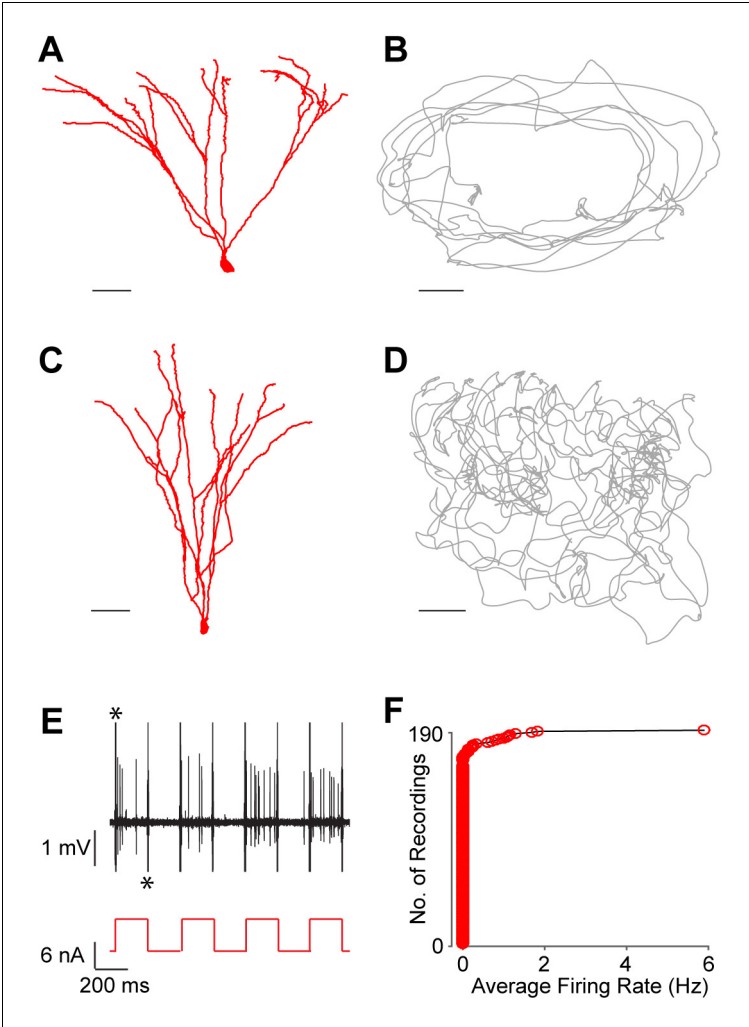

**Figure 2.** Silent granule cells during spatial exploration. (**A** and **C**) Reconstruction of the dendritic morphology of two silent, morphologically mature GCs recorded during freely moving behavior (cell ids 924 and 991, respectively). Scale bars=50 µm. (**B** and **D**) Trajectory of a rat (grey) running on an O-shaped arena (**B**) or on an open-fieldarena (**D**). The two recordings correspond to the neurons in (**A** and **C**), respectively. Scale bars=10 cm. (**E**) Unlike extracellular recordings, juxtacellular sampling is not biased towards active cells, as silent neurons can also be recorded and their presence confirmed by current injection. This is shown here for the silent cell in (**C**): action potential firing (black, top) induced by squared pulses of injected current (red, bottom). These pulses were delivered at the end of the freely moving recording session to confirm the presence of the silent celland to label the cell (**C**). Asterisks indicate stimulation artifacts (truncated for display purposes). (**F**) Cumulative plot showing the firing-rate distribution within the GC layer. Each red circle represents one neuron, sampled juxtacellularly within the GC layer (see Materials and methods for details). Note the large proportion of silent neurons (163 out of 190) compared to that of active cells; this is likely to be an underestimate of the true silent proportion (see Results and Materials and methods for details).

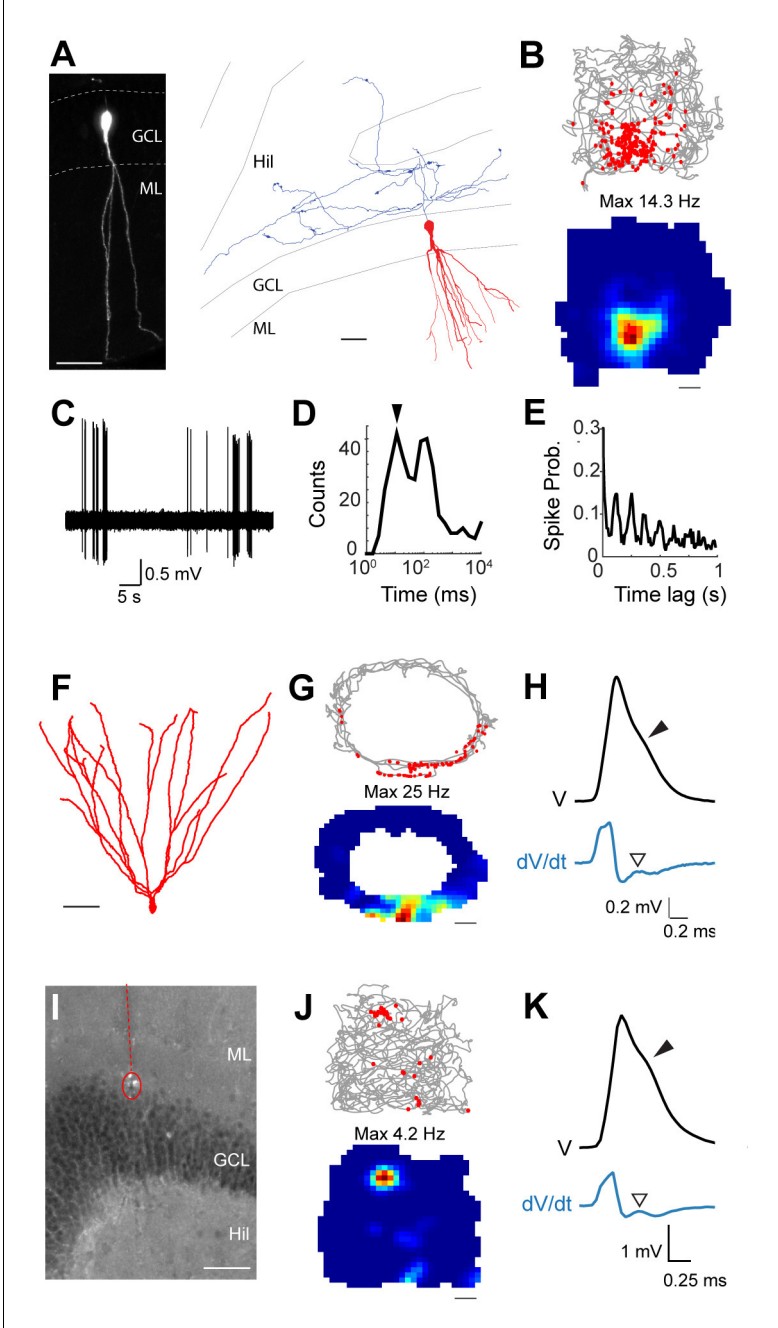

**Figure 3.** Spatial and temporal firing properties of identified active granule cells. (**A**) Left, single-plane confocal image showing a morphologically mature GC (cell id 103) located in the infrapyramidal blade, recorded and labeled in a freely moving animal. Right, reconstruction of dendritic (red) and axonal (blue) morphologies; only the axon extending within the hilus (Hil) is shown. Close anatomical proximity (< 1 μm; compatible with putative synaptic contacts) was observed between mossy boutons and large hilar neurons (4 out of 13 analyzed boutons; not shown), suggesting that hilar neurons might integrate spatial signals from upstream GCs. Scale bars=50 μm. GCL=granule cell layer; ML=molecular layer. (**B**) Spike-trajectory plot (top) and rate map (bottom) for the neuron shown in (**A**), recorded in a square arena. Scale bar=10 cm. (**C**) Representative high-pass filtered spike trace, recorded during free behavior from the neuron in (**A**). (**D**) Interspike interval (ISI) distribution for the neuron in (**A**). Note the early peak (arrowhead) representing short ISI within bursts. (**E**) Spike-autocorrelogram for the neuron in (**A**). Note the prominent rhythmicity in the theta-frequency range (~8 Hz during awake behavior). (**F**) Reconstruction of the dendritic morphology of an active morphologically mature GC (cell id 104) recorded during freely moving behavior. Scale bar=50 μm. (**G**) Spike-trajectory plot (top) and rate map (bottom) for the neuron shown in (**F**)

*Figure 3 continued on next page*

*Figure 3 continued*

recorded on an O-shaped linear arena. Scale bar=10 cm. (H) Normalized average spike waveform (black) of the neuron shown in (F), with corresponding first derivative (blue). Black arrowhead indicates the spike-shoulder and white arrowhead the local peak in the first derivative. The first derivative is scaled up for illustration purposes. (I) Fluorescence micrograph showing a histologically verified electrode track (red dotted line) and recordings site (dotted circle), which was localized to the superficial portion of the GC layer (end of the track). Note the presence of small cell debris at the labeling site (dotted circle). (J) Spike-trajectory plot (top) and rate map (bottom) for the recording location shown in (I) (putative GC recording, cell id 80) during exploration of a square open arena. Scale bar= 10 cm. (K) Same as in (H), but for the putative GC recording in (I).

The following figure supplement is available for figure 3:

**Figure supplement 1.** Spatial firing activity of identified and putative GCs.

compartment of seven silent and six active GCs (all of which were classified as morphologically mature; see *Figure 4A,B*, *Figure 4—figure supplement 1* and criteria in Materials and methods), which were selected for the morphological analysis on the basis of their high-quality filling (see *Figure 4—figure supplement 2B–D* and Materials and methods). We then trained a logistic regression classifier enforcing different levels of sparsity in the weights with different primary morphological metrics (total dendritic length, total length of dendritic branch orders, number of primary dendrites and number of dendritic endings) as features (*Figure 4C* and *Figure 4—source data 1*). The classifier was able to distinguish active and silent cells with good accuracy independent of the enforced level of sparsity (leave-one-out cross-validation accuracy ~85%, 11 out of 13 cells classified correctly). This classification accuracy was significantly higher than chance (p=0.03, permutation test, 500 runs, see Materials and methods). For lower levels of sparsity, the classifier used a more distributed weight profile, whereas for higher levels, the weight used was more concentrated on individual predictors (*Figure 4D*). Overall, longer high-order dendrites (5 and 6 orders) were most predictive of active cells, whereas shorter low-order dendrites (2 and 3 orders) together with a higher number of primary dendrites were most predictive of silent cells (*Figure 4D*). Notably, the morphological differences between active and silent neurons (*Figure 4A,B*) were unlikely to have resulted from methodological or procedural biases (see *Figure 4—figure supplement 2* and Materials and methods for more details). This indicates that in our dataset of identified neurons (n=13), dendritic architectures can be predictive of GC activity during behavior (see also *Figure 4—source data 1*).

## Discussion

Sparse coding in the DG has been postulated to subserve 'pattern separation', a key function during memory processing (*McNaughton and Morris, 1987*; *Leutgeb et al., 2007*; *Myers and Scharfman, 2009*; *Neunuebel and Knierim, 2014*). Here we show that indeed the large majority of juxtacellularly sampled neurons within the GC layer are silent during exploration. The small fraction of active GCs (*Figure 2F*) displayed characteristic juxtacellular spike waveforms, characterized by the presence of 'shoulders' following the positive action potential peak (*Figure 1C,E*). Although, at present, we ignore the relative contributions of capacitive and/or ionic mechanisms to spike-shoulders, they served as a reliable correlate of GC identity. If such waveform signatures can also be resolved extracellularly – which remains to be demonstrated – they could be instrumental for future classification of tetrode-recorded DG units. We also provide direct anatomical evidence that morphologically mature GCs can display spatial firing, which in most cases occurred at single spatial locations (*Figure 3—figure supplement 1*). Although this is consistent with previous work (*Leutgeb et al., 2007*; *Neunuebel and Knierim, 2012*), the limited recording durations and spatial sampling prevent rigorous comparison with previous extracellular studies. Recordings in large open-field environments will be required to resolve whether different firing patterns (single versus multiple firing fields) are contributed by distinct cell types within the DG (*Neunuebel and Knierim, 2012*).

Previous work has demonstrated that cell intrinsic and synaptic mechanisms can be major determinants of single-cell activity in sparsely active networks (*Crochet et al., 2011*; *Epsztein et al.,*

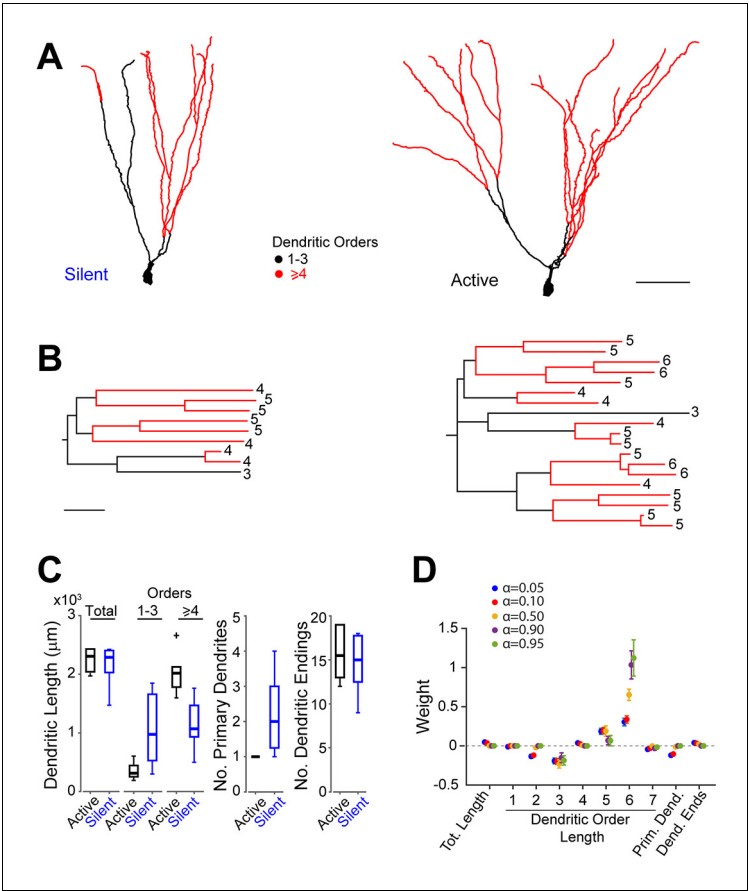

**Figure 4.** Morphological analysis of identified active and silent granule cells. (**A**) Somatodendritic reconstructions of silent (left, cell id 993) and active (right, cell id 102) GCs, recorded in freely moving rats. Dendritic branch orders 1–3 are indicated in black, while high-order branches (≥ 4) are indicated in red. Scale bar=50 μm. (**B**) Dendrograms for the cells in (**A**). The branch orders of the terminal dendritic tips are indicated. Color codes same as in (**A**). Scale bar=50 μm. (**C**) Box-plots comparing primary dendritic parameters (active GCs, n=6; silent GCs, n=7). Whiskers represent 1.5 IQR. Outliers are shown as crosses. (**D**) Weights of logistic regression classifier (mean ± SEM over cross-validation folds) used for classifying GCs as active or silent for different levels of sparseness (α=0.05 corresponds to dense weights, α=0.95 corresponds to sparse weights).

The following source data, source code and figure supplements are available for figure 3:

**Source code 1.** Classification analysis.
**Source data 1.** Electrophysiological and morphological parameters of the identified GCs.
**Figure supplement 1.** Morphological reconstructions of GCs recorded and labeled in freely moving rats.
**Figure supplement 2.** Distribution of recordings across animals and quantification of single-cell labeling quality.

*2011*). Our data indicate that the dendritic architecture of mature GCs can be predictive of in-vivo spiking activity (*Figure 4D*), thus providing a possible structure–function scheme for active and silent neurons. We note, however, that in our limited dataset of identified GCs (n=13) structure–function relationships are subtle, as they derive from the combined analysis of multiple structural parameters (*Figure 4D*). Future work – possibly involving targeted manipulations of dendritic structures – will be required to resolve the causal contribution of GC dendritic architectures to spatial representations.

## Materials and methods

### Juxtacellular recordings

Experimental procedures for juxtacellular recordings, signal acquisition and processing, and animal tracking were essentially performed as recently described (*Tang et al., 2014a*; *Diamantaki et al., 2016*). The electrode solution contained standard Ringer solution: 135 mM NaCl, 5.4 mM KCl, 5 mM HEPES, 1.8 mM $CaCl_2$ and 1 mM $MgCl_2$ (pH is adjusted to 7.2). In a subset of recordings, neurobiotin (1.5–3%; Vector Laboratories, Burlingame, CA) or biocytin (1.5–3%; Sigma-Aldrich, St Louis, MO) was added to the electrode solution. Osmolarity was adjusted to 290-320 mOsm. Sample sizes were estimated on the basis of previously published data using similar procedures (*Ray et al., 2014*; *Tang et al., 2014b*; *Tang et al., 2015*).

Juxtacellular recordings in anesthetized Wistar rats (n=11) were performed under ketamine/xylazine/urethane anesthesia, essentially as previously described (*Klausberger et al., 2003*; *Ray et al., 2014*). Juxtacellular recordings in awake, head-fixed Wistar rats (n=4) were performed as previously described (*Houweling and Brecht, 2008*; *Doron et al., 2014*; *Tang et al., 2014a*). Briefly, animals were pre-implanted with a metal post and a recording chamber under ketamine/xylazine anesthesia, and a craniotomy was performed at the coordinates for targeting the dorsal DG (3.5–4 mm posterior and 1.5–2 mm lateral from bregma). After a recovery period (~3–5 days), animals were slowly habituated to head-fixation. After successful habituation, and before juxtacellular recordings, mapping experiments with low resistance electrodes (0.5–1 MΩ) were performed to estimate the location of the DG precisely. Electrode penetrations were adjusted in order to target the DG blades and the crest. Juxtacellular recordings in freely moving Wistar rats (n=60) were performed as previously described (*Tang et al., 2014a*; *Diamantaki et al., 2016*). Implantation and mapping procedures were similar to those described above for the head-fixed preparation. After recovery from the surgery, animals were slowly habituated to the head-fixation and to collect chocolate/food pellets, as recently described (*Diamantaki et al., 2016*). Training and recording was performed in three types of arenas: a square maze (70 × 70 cm) with 25 cm high walls, a wall-less maze (55 × 65 cm) and an elevated, O-shaped, linear maze (70 × 50 cm, 14 cm wide path). All animals underwent the same training prior to the experiments (> 3 days, > 9 total sessions). 67 recordings from silent neurons in the O-shaped linear maze come from our previous work (*Diamantaki et al., 2016*), and 33 of these neurons were recorded during the first 3 days of exposure to a novel environment; as the fraction of active/silent neurons did not change significantly as a function of habituation (active/silent neurons during the first 3 days of habituation=9/42; active/silent neurons after > 3 days of habituation=18/121; p=0.48, Fisher's Exact Test), these recordings were pooled and included in *Figure 2F* for estimating the proportion of active/silent neurons within the GC layer.

Juxtacellular labeling was performed by using standard labeling protocols (*Pinault, 1996*) and modified procedures, which consisted of rapidly breaking the dielectric membrane resistance with short (1–2 ms) 'buzz-like' current pulses (using the built-in buzz module of the ELC-X03 amplifier, NPI Electronic, Tamm, Germany), which provided rapid access to cell entrainment by juxtacellular current injection, i.e. 200 ms square current pulses, 1–10 nA (*Pinault, 1996*). At relatively high concentrations of neurobiotin or biocytin (i.e. 2.5–3%), short access to the cell's interior was occasionally sufficient for recovering labeled neurons. The same labeling procedures were applied to active and silent neurons, which resulted in similar success rates of juxtacellular filling (see Results). Recordings (or portions of recordings) in which cellular damage was observed – e.g. spike-shape broadening, increase in firing rate accompanied by negative DC-shifts of the juxtacellular voltage signal, as described in (*Pinault, 1996*; *Herfst et al., 2012*) – were excluded from the analysis.

The juxtacellular voltage signal was acquired via an ELC miniature headstage (NPI Electronic), and an ELC-03XS amplifier (NPI Electronic), sampled at 20–50 kHz by a LIH 1600 data-acquisition interface (HEKA Electronic, Lambrecht/Pfalz, Germany) under the control of PatchMaster 2.20 software (HEKA Electronic) or Spike2 v8.02 software and the POWER1401-3 data-acquisition interface (CED, Cambridge, UK). The location of the animal was tracked using two LEDs (red and blue with in-between distance of 3.5 cm) mounted on the rat's head. Animal tracking was performed by acquiring a video (25 Hz frame rate) with the IC Capture Software (The Imaging Source, Bremen, Germany).

## Targeting of the dentate gyrus and dataset of juxtacellular recordings

Electrode penetrations into the DG were accompanied by electrophysiological features, which could be used to estimate the electrode location relative to the layered structure of the DG. Entry into the DG (i.e. the molecular layer) was always associated with a drastic increase in field-potential gamma oscillatory activity during exploratory behavior or urethane/ketamine anesthesia. The beginning of the GC layer could always be easily mapped because the entry into the densely packed cell layer was associated with an increase in the electrode resistance (which was indicative of cell-contact). Multiple juxtacellular hits/recordings were typically established within the layer upon progressively advancing the electrode. The large majority of sampled neurons were silent (see *Figure 2F* and below). Entry into the hilar region was associated with a drop in the electrode resistance, which typically returned to its initial value. Thus electrode resistance, LFP gamma oscillations and juxtacellular hit-rates served as reliable correlates of the electrode location within the GC layer. Indeed, cell identification or anatomical verification of recording sites were in agreement with the expected electrode location.

During the search procedure in freely moving animals, neurons were sampled while rats were exploring the arena by advancing the electrode in small steps (~2–5 µm). An increase in the electrode resistance was considered indicative of cell contact; if spikes were not observed during free-behavior, the silent neurons were either labeled (as in *Figure 2E*) or discarded by further advancing the electrode. In this respect, our search procedures can be considered 'blind'; however, not all silent 'hits' within the GC layer – which were most likely to have resulted from silent neurons – were maintained for a sufficient period of time (> 60 s) to be included in the analysis. This was primarily due to the fact that, for recording spiking GCs, silent 'hits' were often discarded after brief sampling of the arena and the electrode further advanced within the GC layer. In addition to this 'search-bias' towards active neurons, it cannot be formally excluded that other biases – for example, intrinsic to the juxtacellular sampling methods – might have occurred. Nevertheless, our analysis indicates that the dataset of reconstructed neurons (*Figure 4—figure supplement 1*) is likely to represent a random subset of active and silent GCs (see *Figure 4—figure supplement 2A* and Results), and that our morphological results are unlikely to be accounted for by procedural or sampling biases (see also *Figure 4—figure supplement 2*).

Recordings were classified as 'identified' if at least a soma and/or portions of the dendritic tree were recovered (see *Burgalossi et al., 2011*). 'Non-identified' recordings refer to cases in which cell identification by juxtacellular labeling failed, or in which the recordings were lost before labeling could be attempted. Non-identified recordings were classified as 'putative' GCs if: (i) the recording location was assigned the GC layer (based on electrophysiological signatures and/or anatomical verification of the recording sites), (ii) spike-shoulders could be detected (as in *Figure 1—figure supplement 1A*), and (iii) the recording was assigned to the GC class by the neural network classifier (see details in Analysis of Electrophysiology Data). These procedures were conceptually similar to the classification approach developed for layer 2 neurons of the medial entorhinal cortex (*Tang et al., 2014b*) and parasubicular neurons (*Tang et al., 2016*).

The dataset of recordings in anesthetized animals is as follows: from a total of 20 recordings established in the DG, 10 were from identified GCs. Seven recordings were from identified hilar neurons, one from an identified CA3c cell, and one from an identified fast-spiking cell. All included recordings from anesthetized animals were identified.

The dataset of recordings in head-fixed, awake rats is as follows: from a total of nine recordings established in the DG, one was an identified GC. Three recordings were from identified hilar neurons and five from fast-spiking cells. All included recordings from awake head-fixed animals were identified, with the exception of the five neurons classified as fast-spiking neurons based on standard electrophysiological criteria (see details in Analysis of Electrophysiology Data).

The dataset of recordings in freely moving animals includes 190 recordings which were assigned to the GC layer. Recordings from active neurons which were not identified and did not display spike-shoulders (see Analysis of electrophysiology data) were not included in the present study. Only recordings lasting > 60 s were included in the analysis; this threshold, which sets the upper limit on the possible firing rate of silent cells to 0.016 Hz (i.e. < 1 spike in 60 s), was used to define active and silent recordings as those where more than 1 spike or where less than 1 spike in 60 s was fired, respectively. Of the 190 recordings assigned to the GC layer, 163 were from silent neurons (n=161

with firing rate of 0 Hz; n=2 with firing rates < 0.016 Hz; 8 identified). (We note that this proportion of active neurons is likely to represent an overestimate, as silent juxtacellular 'hits' that were shorter than 60 s and were likely to have come from silent neurons, were routinely discarded and not included in the analysis.) In all silent recordings, the presence of a silent neuron was verified either by juxtacellular labeling (as in *Figure 2E*) or by the presence of 'end spikes' upon recording loss. Twenty-seven recordings were from active GCs (9 identified and 18 classified as putative GCs – these 27 recordings were equally distributed between the different types of arenas: 11 were performed in two-dimensional arenas and 16 on the O-shaped circular maze). Eight recordings were from hilar neurons (seven identified and one putative), six from identified CA3c cells and four from fast-spiking cells (one identified and three putative).

All GCs recorded and identified in freely moving animals, for which morphology could be assessed (n=13; *Figure 4—figure supplement 1* and *Figure 4—source data 1*), were classified as morphologically mature neurons according to previously established criteria; for example, based on their location within the GC layer, the absence of basal dendrites, the complex cone-shape of the dendritic arbor and, in the few cases where axons were recovered (e.g. *Figure 3A*), the presence of long-range axons and mossy boutons (*Ambrogini et al., 2004*; *Schmidt-Hieber et al., 2004*; *Espósito et al., 2005*; *Zhao et al., 2006*; *Pernía-Andrade and Jonas, 2014*). The few cells located within the lower third of the GC layer (n=3; of which two were silent and one active; see *Figure 4— source data 1*) were tested for doublecortin expression and were negative (not shown). This result is in line with the expected small proportion (~5%) of morphologically immature GCs relative to the total GC population in young adult rats (*Cameron and McKay, 2001*; *Gould, 2006*).

## Analysis of electrophysiology data

The animal's speed was calculated on the basis of smoothed X and Y coordinates of the tracking (averaged across a 600 ms rectangular sliding window). Statistical significance was assessed by the two-sided Mann-Whitney nonparametric test with 95% confidence intervals. Data are presented as mean ± standard deviation unless otherwise indicated.

The 'spike-shoulder' analysis was performed on the average spike traces. The presence of 'spike shoulders', occurring after the positive peak of the juxtacellular spike, was determined using a moving average approach as follows. First a search window on the spike waveform was defined between the spike-peak and the spike-offset (10% of the spike peak). A moving average of the voltage signal was computed (using a 0.25 ms sliding window), and the spike trace within the search window was compared to its moving average. The occurrence of an intersection between these two was used as an identifier for a 'shoulder' in the average spike trace. If a shoulder was identified, the half-width of the local maximum on the corresponding portion of the first derivative of the spike trace was used as a quantifier for the prominence of the 'spike shoulder' (as shown in *Figure 1G*). To eliminate noise in measuring the local maximum, both the spike trace and its first derivative were smoothed (averaged across a 200 ms rectangular sliding window).

The dataset of juxtacellular recordings for 'spike-shape analysis' included all identified active neurons (total of n=47 neurons): identified GCs (n=21), hilar cells (n=17), CA3c cells (n=7) and fast-spiking cells (n=2). The category of fast-spiking neurons contained identified cells (n=2) and non-identified recordings (n=8), which were classified as fast-spiking on the basis of characteristic electrophysiological features (i.e. firing rates > 10Hz and/or narrow spike widths, i.e. < 0.2 ms spike half-widths). One neuron, located below the GC layer (asterisk in *Figure 1D*) displayed spike-shoulders, but its morphology could not be assessed because only the soma was recovered; this neuron was, however, classified as a putative excitatory cell (positive for GluR2/3 expression; not shown) and, unlike mossy cells or displaced CA3c neurons, it displayed a small soma (diameter=11 μm). Based on these observations and previous criteria (*Amaral and Woodward, 1977*; *Martí-Subirana et al., 1986*; *Tóth and Freund, 1992*; *Scharfman et al., 2007*; *Szabadics et al., 2010*; *Scharfman and Pierce, 2012*), this neuron was classified as an ectopic GC. We note, however, that including this neuron in the hilar class did not significantly affect the classification accuracy of the neural network classifier and all conclusions from the present work remained unaffected (not shown).

To test whether unidentified juxtacellular recordings could be assigned to the GC class on the basis of their average spike waveform, we used a neural-network classifier that was trained on the average spike waveforms of all identified active neurons (n=47 neurons; 'spike shape analysis' dataset, as above). The neural-network classifier was created using the built-in functions of the

MATLAB Neural Networks Toolbox. The hidden layer size was set to ten and training was stopped when more than six validation errors occurred. To train the neural-network classifier, the Levenberg-Marquardt backpropagation and a leave-one-out cross validation scheme was used. To balance the different class sizes, we used the ADASYN (adaptive synthetic sampling approach for imbalanced learning) algorithm. This algorithm creates new samples from the minority class using linear interpolation between existing minority class examples (*Haibo et al., 2008*). GCs were correctly classified on the basis of spike-shape features with high accuracy (~89%) and with a low rate of false-positives (2/47; ~4%) and false-negatives (3/47; ~6%). All neurons that were classified as 'putative' GCs (n=18) were correctly classified as GCs by the neural network classifier (see also 'Targeting of the dentate gyrus and dataset of juxtacellular recordings' above). High classification accuracy (~87%) was also obtained by training the classifier only on the repolarization phase of the average spikes (i.e. where shoulders occur), indicating that the presence/absence of spike-shoulders is a strong determinant of the classifier's performance. In line with the conclusions from the identified dataset (*Figure 1D–F*), the classification results provide additional evidence for a correlation between spike-shoulders and GC identity.

## Rate maps and analysis of spatial firing

Animal positional coordinates were extracted from the video (format: AVI, frame rate: 25 Hz) by custom-made software written in MatLab. After calculating the LED positions frame by frame, the position of the rat was defined as the midpoint between two head-mounted LEDs. A running speed threshold (> 1 cm/s) was applied for isolating periods of rest from active movement. Color-coded firing maps were plotted. For these, space was discretized into pixels of 2.5 cm x 2.5 cm, for which the occupancy $z$ of a given pixel $x$ was calculated as:

$$z(x) = \sum_t w(|x - x_t|)\Delta t$$

where $x_t$ is the position of the rat at time $t$, $\Delta t$ is the inter-frame interval, and $w$ is a Gaussian smoothing kernel with $\sigma$ = 2.5 cm.

Then, the firing rate $r$ was calculated as:

$$r(x) = \frac{\sum_i w(|x - x_i|)}{z}$$

where $x_i$ is the position of the rat when spike $i$ was fired. The firing rate of pixels whose occupancy $z$ was less than 20 ms was considered unreliable and not shown. Spatial coverage was defined as the fraction of visited pixels (bins) in the arena . Inclusion criteria for spatial analysis were as follows: (i) spatial coverage > 70% for the square arenas or $\geq$ two laps for the linear arena; (ii) average firing rate > 0.1 Hz and a total number of spikes > 25. Sixteen out of 27 GC recordings met these criteria, and were included in the spatial analysis. The generally low firing rates and total number of spikes prevented rigorous quantification of spatial modulation by statistical criteria; in line with observations from previous work (*Jung and McNaughton, 1993*), we classified recordings as spatially selective if the spikes were spatially clustered and the corresponding place fields accounted for < 20% of the total visited pixels. Spatial information content was calculated as in *Bjerknes et al. (2014)*. To calculate the number of place fields, a firing field was defined from the rate map, and considered as at least 12 contiguous pixels with average firing rate exceeding 20% of the peak firing rate. The 'in-field firing rate' (*Figure 4—source data 1*) was calculated as the average firing rate of the place field pixels, and the 'out-of-field firing rate' as the average firing rate of the pixels outside the place fields. In eight recordings, a single firing field could be observed (see *Figure 3*), where as in three cases, an additional field could be detected. In the remaining five recordings, firing was not spatially selective. We note that the limited durations of our recordings and the generally lower spatial coverage prevents direct comparison with previous tetrode studies (*Jung and McNaughton, 1993*; *Leutgeb et al., 2007*; *Neunuebel and Knierim, 2012*); our findings are, however, largely in agreement with previous literature, which indicates that most spatially selective units sampled within (or near) the GC layer contribute single firing fields (*Leutgeb et al., 2007*; *Neunuebel and Knierim, 2012*).

## Histochemistry and cell reconstruction

At the end of each recording, the animal was euthanized with an overdose of pentobarbital and perfused transcardially with 0.1 M phosphate-buffered saline followed by a 4% paraformaldehyde solution. Brains were cut on a vibratome to obtain 50–70 μm thick coronal sections. To reveal the morphology of juxtacellularly labeled cells, brain slices were processed with streptavidin-546 or 488 (Life Technologies, Carlsbad, CA) as previously described (*Tang et al., 2014a*). Immunohistochemical stainings for doublecortin (C-18, Santa Cruz Biotechnology, Dallas, Texas) were performed as previously described on free-floating sections (*Ray et al., 2014*). After fluorescence images were acquired, the neurobiotin/biocytin staining was converted into a dark DAB reaction product. All reconstructed active and silent neurons (n=13, *Figure 4—figure supplement 1*) underwent the same histological processing and were processed following the $Ni^{2+}$-DAB enhancement protocol (*Klausberger et al., 2003*). Neuronal reconstructions were performed manually on DAB-converted specimens with the Neurolucida software (MBF Bioscience, Williston, VT) and displayed as two-dimensional projections.

## Analysis of morphological data

Morphometric analysis of the reconstructed neurons was performed with the Neurolucida software. Neuronal reconstructions were performed blind to the electrophysiological properties of the cells. For cell id 993, the soma was occluded by a black deposit on the brain section, and its position was manually interpolated at the termination point of the primary dendrite. Close-up magnifications of the soma and proximal dendrites in *Figure 4—figure supplement 1* have, in a few cases, been rotated along the longitudinal axis to allow optimal display of the primary dendrites. The quality of cellular filling and labeling was estimated by quantifying the gray levels of DAB-stained neurons with ImageJ (*Schneider et al., 2012*). For all identified neurons (n=13; *Figure 4—figure supplement 1*), high-magnification pictures (100x oil-immersion objective) were obtained under identical settings (e.g. illumination and camera exposure) with a light microscope (BX-53, Olympus). A region of interest was defined on proximal and distal dendrites, and the normalized gray scale values (max gray value – background level) were taken as a correlate of labeling intensities. All proximal dendritic segments were within 20 μm of the soma, while distal dendritic segments were ≥ 200 μm from the soma. We note that in both groups (active and silent) of reconstructed neurons, it cannot be assured whether - even in the best-filled examples - dendritic morphologies are full and complete. Nevertheless, our analysis indicates that labeling quality and filling efficiency were very similar between active and silent neurons (*Figure 4—figure supplement 2B–D* and 'control experiments' below). Labeling efficiencies for silent and active neurons were estimated as the proportion of filled neurons (n=6 active and n=7 silent neurons; *Figure 4—figure supplement 1* and *Figure 4—source data 1*) over all labeling attempts. Dendritic lengths were not compensated by tissue shrinkage. Dendritic order ($n^{th}$) was defined as the portion of the dendritic tree between the (n−1) and (n) branching nodes, or the dendritic end. The only exception was the first-order dendrites, which were estimated as the distance between the soma center and the first branching node. We found this measure to be more consistent across repeated measurements and experimenters, whereas the distance from the soma edge to the first branching node is more dependent on the shape (more or less elongated) of the somatic contour. Estimating primary dendritic length as the distance between the soma edge and the first branching node (see dendritic segments in 'blue' in *Figure 4—figure supplement 1*) did not affect classification accuracy (not shown). The 'soma location' within the GC layer (see *Figure 4—source data 1*) was estimated as the relative position with respect to the anatomical borders of the GC layer, with '0' representing the deep border – close to the hilus – and '1' the superficial border – close to the molecular layer. The 'branching index' was defined as the number of dendritic endings divided by the number of primary dendrites. The 'complexity index' was computed, as in previous work (*Pillai et al., 2012*), according to the following equation: ($\Sigma$ branch tip orders + number of branch tips)×(total dendritic length/total number of primary dendrites). To estimate the total number of spines (see *Figure 4—source data 1*), we quantified spine densities in GCs (n=7) by manually counting the number of spines on 10 μm dendritic segments, and then dividing it by the total segment length; at least three dendritic segments were counted per branch order, and averaged. Spine densities were within the range reported in the literature (*Desmond and Levy, 1985*; *Hama et al., 1989*; *Trommald and Hulleberg, 1997*; *Vuksic et al.,*

*2008*; *Freiman et al., 2011*); however, in the present study, densities were not compensated for hidden spines and so absolute values represent an underestimate. Since spine densities increased as a function of dendritic order (not shown; see also *Stone et al. (2011)*), the total number of spines (*Figure 4—source data 1*) was estimated as the sum, over all dendritic orders, of the spine densities multiplied by the total dendritic length of each order.

## Morphological analysis: control experiments

To determine whether the morphological differences between active and silent neurons could have resulted from procedural or methodological biases, we preformed the following controls. First, the efficiency of juxtacellular filling was very similar between active and silent neurons (active, 6/14 labeling attempts, ~42%; silent, 7/19 labeling attempts, ~37%). Second, no apparent bias was observed in the distribution of active and silent neurons across animals (*Figure 4—figure supplement 2A*), and no correlation was observed between in-vivo activity and soma location within the GC layer or the laminar location of the cells (suprapyramidal versus infrapyramidal blade) (see *Figure 4—source data 1*). Third, the quality of cell filling and dendritic labeling was similar between active and silent neurons. This was assessed by comparing: (i) the staining intensity at distal dendritic locations (*Figure 4—figure supplement 2B,C*); and (ii) the distribution of dendritic endings as a function of distance from the soma (*Figure 4—figure supplement 2D*). We reasoned that incompletely filled neurons should display lower labeling intensities and dendritic endings that are located closer to the soma; these parameters (together with total dendritic lengths, see *Figure 4—source data 1*) were not significantly different between active and silent neurons (see *Figure 4—figure supplement 2B–D*). Moreover, dendritic complexity was not correlated to the labeling/filling quality (p=0.32; n=13). Altogether,these analyses indicate that our identified neurons are likely to represent a random sample of active and silent GCs, and that labeling biases are unlikely to account for the differences in morphological parameters between the two classes.

## Sparse logistic regression classifier

To build an automatic cell-type classifier that uses morphological parameters, we trained an L1-regularized logistic regression classifier (*Friedman et al., 2010*) using nested cross-validation (*Cadwell et al., 2016*). In the outer cross-validation loop, we iterated over each individual cell, evaluating the prediction for that cell with a model trained on all but that cell ('leave-one-out cross-validation'). We used primary dendritic measures (the total dendritic length, the length of dendritic orders 1–7, the number of primary dendrites and the number of dendritic endings) as features (10 in total; see *Figure 4D*). Training on additional derived measures (such as 'complexity index' or 'branching index'; see *Figure 4—source data 1*) did not improve classification performance (not shown; see *Figure 4—source code 1*).

Cross-validation in the inner loop was used to select the optimal amount of regularization. The evaluated model is selected on the basis of the the 1-SE rule to prevent overfitting (*Friedman et al., 2010*). All variables were z-scored before training the classifier. We used the elastic-net algorithm with $\alpha$=0.05, 0.1, 0.5, 0.9 and 0.95, regularizing the weights with a mix of L2 and L1 regularization; this resulted in a denser weight vector for small values of $\alpha$ and a sparse weight vector for high values. The elastic-net penalty is particularly well suited for coping with highly correlated predictors as is the case here (*Friedman et al., 2010*). The choice of $\alpha$ did not affect the classification performance (~85%). We used the implementation provided by lassoglm in MATLAB with a binomial output distribution. We report the mean weight with SEM across cross-validation runs. To test whether the classification performance is significantly different than chance, we performed a permutation test. To this end, we shuffled the labels of all cells and repeated the entire cross-validation and classification procedure to obtain a null distribution of classification performances (500 runs, only for $\alpha$=0.1). The p-value for the classification performance on the actual data is computed as the fraction of runs resulting in a classification performance larger or equal to the one observed in the data. Code for running this analysis is available with the paper (*Figure 4—source code 1*).

## Acknowledgements

This work was supported by the Werner Reichardt Centre for Integrative Neuroscience (CIN) at the Eberhard Karls University of Tübingen (CIN is an Excellence Cluster funded by the Deutsche

Forschungsgemeinschaft within the framework of the Excellence Initiative EXC 307) and by the the Bernstein Award from the German Ministry of Science and Education (BMBF) to PB (FKZ 01GQ1601). We thank Alexandra Eritja for excellent assistance with anatomy experiments, and Nima Ghorbani for contributing to software development. We thank James Knierim and Xioajing Chen for helpful discussions.

## Additional information

### Funding

| Funder | Grant reference number | Author |
| --- | --- | --- |
| Deutsche Forschungsge-meinschaft | EXC 307 | Maria Diamantaki<br>Markus Frey<br>Philipp Berens<br>Patricia Preston-Ferrer<br>Andrea Burgalossi |
| Bundesministerium für Bildung und Forschung | FKZ 01GQ1601 | Philipp Berens |

The funders had no role in study design, data collection and interpretation, or the decision to submit the work for publication.

### Author contributions

MD, Acquisition of data, Analysis and interpretation of data, Drafting or revising the article; MF, PB, Analysis and interpretation of data, Drafting or revising the article; PP-F, AB, Conception and design, Acquisition of data, Analysis and interpretation of data, Drafting or revising the article

### Author ORCIDs

Philipp Berens, http://orcid.org/0000-0002-0199-4727
Andrea Burgalossi, http://orcid.org/0000-0003-0039-3599

### Ethics

Animal experimentation: All experimental procedures were performed according to the German guidelines on animal welfare and approved by the local institution in charge of experiments using animals (Regierungspraesidium Tuebingen, permit numbers CIN2/14, CIN/5/14 and CIN/814).

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
