## [Decision Letter]

Thank you for submitting your article "Structural determinants of granule cell activity in the dentate gyrus of freely-moving rats" for consideration by *eLife*. Your article has been favorably evaluated by Eve Marder (Senior Editor) and three reviewers, one of whom, Karel Svoboda (Reviewer #1), is a member of our Board of Reviewing Editors, and another one is Matthew F Nolan (Reviewer #2).

The reviewers have discussed the reviews with one another and the Reviewing Editor has drafted this decision to help you prepare a revised submission.

Summary:

This manuscript describes loose-seal cell-attached recordings from hippocampal granule cells (GCs) in awake rats. We know relatively little about these mysterious cells. Previous experiments have explored granule cell activity using tetrode-style recordings or maps of cFos expression. These methods have suggested that GC activity is sparse – in the sense that only a small subset of GCs are active in any one environment – but both methods are imperfect. Extracellular recordings are biased towards active cells and sample neurons with more membrane area more efficiently. Loose-seal recordings are unbiased in the sense that active and inactive neurons are recorded. The authors find ultra-sparse activity in GCs, providing the strongest evidence to data for sparse coding in the dentate gyrus. Loose-seal recordings in addition allow labeling and reconstruction of a subset of recorded neurons. This provides the additional insight that active neurons are larger and spiny. This is a nice and simple structure-function relationship. These findings of specific structure-function relationships within the (mature) dentate granule cell population are novel and very interesting, and would constitute a valuable contribution to the field.

Essential revisions:

All reviewers have reservations about the strength of the statistical analysis used to support the main conclusions and presentation of the data. We suggest that you take the comments below as a guide to reanalyze the data set.

1) Details on the GC recordings and morphological parameters should be provided. We suggest a supplementary table containing all neurons as separate entries with these data: duration of recording; number of spikes recorded; identity of the animal; duration of the recording; distance covered by the animal; aspects of receptive field shape (i.e. place field attributes), morphological parameters; which analysis and figure the neuron contributed to. This table should allow independent analysis of the dataset. The detailed dendritic structures should be deposited at neuromorph.org or another repository.

2) Given multiple comparisons of dendritic structure were made, it is unclear if corrections for multiple comparisons were made. This is related to the hot topic of 'p-value hacking'. Please make sure to take multiple comparisons into account.

3) The study reports numbers of neurons, treating each neuron as an independent observation. Could the apparent relationship between activity and dendrite morphology instead reflect a subset of animals in which granule cells are more active with more complex dendrites? The possibility of non-independence of observations should be accounted for in the statistical analysis.

4) The description of the classifier and its evaluation is not sufficient. What properties of the spike waveform were used for training? What was the exact composition of the training and test datasets? Is it just the 47 neurons in Figure 1? This is a somewhat small number of samples compared to datasets often used for training classifiers. What was the false positive and false negative rate when testing the classifier? What was the quantitative criterion for shoulder or no shoulder in Figure 1? Figure 1 indicates that the shoulder is variable and not present in all granule cells. Does this have implications for interpretation of the data and what are these?

5) Figure 4. The legend indicates that "the distributions" are statistically different. Is this referring to length, distance, or length data as a function of distance as shown in the figure? The conclusion is based on results of a Mann Whitney U which would usually be used to test for a difference in medians. It is not clear how this was applied here.

The analysis of morphology and its presentation needs to be improved. Assuming that the morphologies of the 7 silent and 6 active granule cells analyzed in detail are complete and representative, the main conclusions of the study are fairly straightforward. Therefore, it is important for the authors to devote more of the manuscript to their morphological methods and clarify their procedures in more detail.

6) The authors write "we reconstructed the somato-dendritic compartment of 7 silent and 6 active GCs, which were selected for the morphological analysis due to their high-quality filling and complete dendritic morphology." What were the criteria for assessing the completeness of the dendritic morphology? Was the labeling protocol standardized in some way between silent and active neurons? For instance, is it possible that it was harder to fill silent cells than active ones? What is the fraction of silent cells for which labeling was attempted whose dendritic tree was completely filled, and how does that compare to the fraction for active cells? If there was any difference in the rate of complete filling, could it be possible that there was a bias in the labeling method that could lead to an apparent difference in the level of branching or complexity? Is it possible, for example, that silent cells were harder to fill and more likely to have somewhat less complete dendritic fills, and thus more likely to appear to have less complex trees?

7) The branching and complexity measures that show a difference between active and silent cells are defined in the methods: "The 'branching index' was defined as the number of dendritic endings divided by the number of primary dendrites. The 'complexity index' was computed as in previous work (Pillai et al., 2012) according to the following equation: (sum of branch tip orders + number of branch tips) x (total dendritic length/total number of primary dendrites)." These measures appear to be particularly sensitive to the number of primary dendrites as it appears in the denominator in both cases. Therefore, the authors should include in the table mentioned above, for each of the 13 neurons with complete dendritic trees, the firing rate in the arena, the branching and complexity values, the total dendritic length, the number of dendritic endings, and the number of primary dendrites. The authors write that there was a non-significant 2x difference in the number of primary dendrites "(the number of primary dendrites (active, 1.1+/-0.4; silent, 2.2+/-1.1; p=0.08)", but the branching index also differed by 2x "(branching index; active, 14.6+/-4.6; silent, 7.8+/-4.7; p=0.026)", though this difference was significant. This suggests that the number of dendritic endings is similar for both silent and active cells. Therefore, is it possible that the difference in the number of primary dendrites could be responsible for differences in activity, instead of the difference in spine number that they suggest as the cause? In comparison, the number of spines differed by a factor of 1.4x "(total number of spines; active, 1802+/-353; silent, 1354+/-257; p=0.035)" and they state "The total number of dendritic spines also showed a significant correlation with firing rates (r=0.63; p=0.019; n=13)". What is the correlation between the number of primary dendrites and firing rate and associated significance?

8) The authors write that "The large majority of sampled neurons were silent (see Figure 2). In order to record from spiking GCs, silent recordings were routinely discarded by further advancing the electrode within the layer." Does this mean that their sample was biased toward more active cells? If so, they should clearly point this out when they state proportions of cells that were active. For instance, they state "Active GCs were thus very sparse, consistent with previous estimates (Jung and McNaughton, 1993; Leutgeb et al., 2007; Neunuebel and Knierim, 2012) and accounted for only ~14% (32 out of 228) of all blindly-sampled neurons within the GC layer." If they biased their search for active cells, then they should remove "blindly-sampled" from this sentence and point out that the 14% value is an overestimate based on their sampling method. In addition, is it possible that any bias in the search toward active cells could influence the morphological results? For instance, is it possible that cells that are active during the search and active during exploration of the arena are not a random sample of the entire set of cells active during exploration? Perhaps these cells have more branching and complex dendritic trees compared to cells that are active during exploration but were not spontaneously active during the cell search procedure? The basic question is whether the sample of 7 silent and 6 active cells analyzed in detail morphologically constituted an unbiased sample of silent cells, and an unbiased sample of active cells, respectively. If the possibility of such a bias cannot be ruled out, yet there is no reason to think there is a bias, the authors could just make a short comment about how they assume their methods have isolated a random sample of silent and active cells.

How long did they wait before discarding silent cells? This defines an upper bound on the possible spike rate of silent cells.

The authors recently showed that they could induce place fields in some granule cells by juxtacellular electrical stimulation (Diamantaki et al. 2016). Does this effect also correlate with dendritic structure? Please discuss.

The paper should be more tightly framed in terms of measurement of sparseness (additional literature should be cited) and structure-function-relationship.

[Editors’ note: a previous version of this study was rejected after a second round of peer review, but the authors submitted for reconsideration. The decision letter after this second round of review is shown below.]

Thank you for submitting your work entitled "Structural correlates of granule cell activity in the dentate gyrus of freely-moving rats" for consideration by *eLife*. Your article has been reviewed by two peer reviewers, and the evaluation has been overseen by a Reviewing Editor (Karel Svoboda) and Eve Marder as the Senior Editor. The following individuals involved in review of your submission have agreed to reveal their identity: Karel Svoboda (Reviewer #1) and Matthew F Nolan (Reviewer #2).

Our decision has been reached after consultation between the Reviewing Editor and the reviewers. Based on these discussions and the individual reviews below, we regret to inform you that your work will not be considered further for publication in *eLife*.

We note that the reviewers remain very positive about the data set and the finding of sparse coding in the dentate gyrus. However, the focus of the manuscript is the link between activity and dendritic structure (the title is 'Structural correlates of granule cell activity in the dentate gyrus of freely-moving rats'). Given this focus all reviewers and the Reviewing Editor remain concerned about the statistical analysis linking activity and dendritic structure. Some commonly used measures of dendritic complexity, such as dendritic length and the number of primary dendrites, showed no significant effect, whereas some less intuitive measures (i.e. 'dendritic complexity') did, but at modest levels of significance. Most of the reported measures are related and thus not independent. There are some indications that the study may be underpowered to make conclusions about dendritic structure. We are not convinced that the results were corrected for multiple comparisons together in an appropriate manner. For these reasons we cannot publish your manuscript in its present form.

However, we would be happy to consider the manuscript again with one of the following major revisions:

1) A more compelling statistical analysis. This could take the form of a rigorous analysis of multiple comparisons. Alternatively, a reanalysis with randomly chosen 50% of the data might also be appropriate.

2) Addition of data that make the conclusions about structure-function relationships stronger.

3) A refocusing on sparse coding in the dentate gyrus.

Reviewer # 1:

The paper is improved in many ways. However, the analysis of dendritic complexity remains suggestive, but not convincing.

Was the study design a priori to test the relationship of spike rate and the specific measure of dendritic complexity? If so, the analysis is appropriate. However, if this was exploratory then the stats are weak. The reasons are as follows.

There is one effect – higher dendritic complexity for higher spike rates. All the positive effects come from the same data and follow from each other; thus there is no independent second measurement. Of course, If one does enough comparisons there is bound to be one that is 'significant' by chance. I don't see a proper analysis of multiple comparisons (this may be hard to do).

Multiple testing and reporting significant results produces a publication bias. In other words, statistically significant findings are reported, increasing the rate of false positives in the literature. This is a prime cause of the poor replication record in psychology, MRI, cancer, and likely systems neuroscience.

Figure 4 seems to have 11 points, not 13. It is hard to believe that the p value is < 0.01

*Reviewer #2:*

The manuscript is substantially improved. However, I'm not convinced the issues related to multiple comparisons have been adequately addressed (points 2 and 5 in the original review).

The investigation of the relationship between activity and morphology requires comparing many parameters. Of course, by chance some of these comparisons will turn out significant at a p < 0.05 level. At the moment this possibility is not accounted for in the interpretation of the data. The challenge here seems to be to introduce an appropriate correction.

Using the Benjamini & Hochberg method to correct the p-values most of the significant differences appear to go away. An exception appears to be the 'Dendritic Length Order' measure. Even here one should perhaps be a little cautious as additional comparisons were made and correcting for them would reduce the significance level further – it is unclear how many comparisons one should correct for.

Given these potential issues, the question is what to do. I appreciate the challenges in obtaining the data, but in a sense this makes it even more important to be rigorous in interpretation. One option might be to carry out additional experiments to try to replicate the results with the analysis focussed on specific planned comparisons. An alternative could be to clearly label the study as exploratory.

Different multiple comparison issues apply to Figure 4. I also don't understand how total dendritic length can be similar between the two groups, but the area of the plots in 4D looks quite different.

Additional comments:

The table only includes a subset of the measurements used for the analysis. It would be more helpful if all measurements are included.

It's not clear why the Abstract reports correlations rather than results of comparisons between active and silent neurons.

The recording duration is longer for active than silent cells. Is this an issue?

*Reviewer #3:*

The authors have adequately addressed my questions and concerns. I have a few remaining comments:

From the table, the numbers that stand out most are the differences in the number of primary dendrites, even though this did not reach significance.

In the Abstract, the authors write "We found that the majority of neurons (163 of 190) were silent during exploration." However, I recommend that the authors remove the numbers since it reads as if it is an unbiased estimate of the fraction of silent neurons, but they confirmed in the response that this is likely to be underestimate due to their search procedure. Instead they could add "vast" in front of "majority" to make their point.

In the Figure 2 legend the authors write "Unlike extracellular recordings, juxtacellular sampling is not biased towards active cells, since silent neurons can also be recorded and their presence confirmed by current injection." This is followed by "Cumulative plot showing the firing rate distribution within the GC layer. Each red circle represents one neuron, sampled juxtacellularly within the GC layer (see Methods for details). Note the large proportion of silent neurons (163 out of 190) compared to active cells." However, as acknowledged by the authors, their sampling of the proportion of active and silent cells is likely to be biased due to their search procedure, where inactive cells were often discarded early during exploration, before the 60-second threshold they used for counting the cells. Because of this, I think this statement and the numbers could confuse the readers into thinking that these numbers are an unbiased estimate, and one that is a better estimate than obtained with other methods. It would be a better estimate if their cell search / counting procedure was unbiased, but apparently it was not. The correct proportion is an important number. The authors should therefore clarify this in the legend as they have done in the main text. For example, they could write in the legend "Note the large proportion of silent neurons (163 out of 190) compared to active cells. Furthermore, this proportion is likely to be an underestimate of the true silent proportion due to the details of the search procedure (see text and methods)."

[Editors’ note: what now follows is the decision letter after the authors submitted for reconsideration.]

Thank you for submitting your article "Sparse activity of identified dentate-gyrus granule cells during spatial exploration" for consideration by *eLife*. Your article has been favorably evaluated by Eve Marder (Senior Editor) and three reviewers, one of whom, Karel Svoboda (Reviewer #1), is a member of our Board of Reviewing Editors, and another one is Matthew F Nolan (Reviewer #2).

The reviewers have discussed the reviews with one another and the Reviewing Editor has drafted this decision to help you prepare a revised submission.

Summary:

This manuscript describes loose-seal cell-attached recordings from hippocampal granule cells (GCs) in awake rats. We know relatively little about these mysterious cells. Previous experiments have explored granule cell activity using tetrode-style recordings or maps of cFos expression. These methods have suggested that GC activity is sparse – in the sense that only a small subset of GCs are active in any one environment – but both methods are imperfect.

Extracellular recordings are biased towards active cells and sample neurons with more membrane area more efficiently. Loose-seal recordings are unbiased in the sense that active and inactive neurons are recorded. The authors find ultra-sparse activity in GCs, settling and quantifying the issue of sparseness in a definitive manner. Loose-seal recordings in addition allow labeling and reconstruction of a subset of recorded neurons.

This is a revised submission, which is more tightly focused on sparse coding in the dentate gyrus. The analysis of structure-function relationships is now done using a classifier with multiple structural parameters considered jointly. The classifier reveals that structure predicts function.

Essential revisions:

Although the classifier can use structure to predict function, the relationship is subtle and can't clearly be boiled down to simple measures (at least given the limited data set) (see also Figure 4—figure supplement 1). It would be good to make this last point clearer (the relevant statement in the Discussion is not accurate and should be revised).

Describe more clearly what a primary, as in 1st order, dendrite compared to a 2nd order dendrite. This is because (1) it appears from the table that this this would have a large effect on distinguishing active and silent cells, not just in terms of the total 1st order length, but also in terms of the total higher order lengths (since a 5th order branch would be a 4th order branch if the 2nd order branch it came from was instead called a 1st order branch), and (2) the table shows that the 1st order branches are generally short. Therefore, in the Figure 4—figure supplement 1 the authors should include beside each neuron a close-up of the somatic region showing the soma, 1st order branch(es), and start of 2nd order branches, ideally with markers to show where the divisions are.

It would be valuable to the community for the authors to include some of the data from the original manuscript in the source file on morphological parameters, such as the "soma location within the GC layer" and the "laminar location of the cells (suprapyramidal versus infrapyramidal blade)."

---

## [Author Response]

*[…] Essential revisions:*

*All reviewers have reservations about the strength of the statistical analysis used to support the main conclusions and presentation of the data. We suggest that you take the comments below as a guide to reanalyze the data set.*

*1) Details on the GC recordings and morphological parameters should be provided. We suggest a supplementary table containing all neurons as separate entries with these data: duration of recording; number of spikes recorded; identity of the animal; duration of the recording; distance covered by the animal; aspects of receptive field shape (i.e. place field attributes), morphological parameters; which analysis and figure the neuron contributed to. This table should allow independent analysis of the dataset. The detailed dendritic structures should be deposited at neuromorph.org or another repository.*

Along with the reviewers’ suggestion, we provide a more comprehensive overview of our dataset by including a table and 2 figure supplements. We have also contacted Giorgio Ascoli for depositing our reconstructed neurons at neuromorph.org, which will be readily uploaded upon acceptance.

We provide a table which includes the major electrophysiological and morphological properties of our reconstructed silent and active neurons ([Supplementary-material SD2-data]). We have also included 2 additional figures, where we show the morphology of all reconstructed neurons (Figure 4—figure supplement 1) and spatial firing activity in our dataset (Figure 3—figure supplement 1). Cell ids are indicated in the corresponding figures and figure legends. These data are referred to in the revised Results (fourth and fifth paragraphs).

*2) Given multiple comparisons of dendritic structure were made, it is unclear if corrections for multiple comparisons were made. This is related to the hot topic of 'p-value hacking'. Please make sure to take multiple comparisons into account.*

Multiple comparisons were taken into account by performing a multivariate ANOVA with two independent groups (silent/active) for comparing dendritic morphological parameters (see [Supplementary-material SD2-data]). Post-hoc pairwise comparisons were assessed with a two-sided Mann-Whitney nonparametric test (95% confidence intervals).

This is specified in the revised Methods (end of last paragraph).

*3) The study reports numbers of neurons, treating each neuron as an independent observation. Could the apparent relationship between activity and dendrite morphology instead reflect a subset of animals in which granule cells are more active with more complex dendrites? The possibility of non-independence of observations should be accounted for in the statistical analysis.*

To address the reviewers’ comment, we now provide the distribution of our recordings across all rats (Figure 4—figure supplement 2). As it can be seen in the histogram, each rat contributed on average a very low number of recordings (average recordings per rat, 3.2 ± 2.7; median = 2). This was primarily due to the fact that, in order to achieve unequivocal cell identification (which is crucial for resolving structure-function relationships) only very few electrode penetrations and labeling attempts (typically one) were performed per animal. In the subset of rats where a larger number of neurons was recorded (see Figure 4—figure supplement 2), labeling was either not attempted, or cell identification could not be determined with certainty. As a consequence of this sparse sampling, the large majority of identified recordings (16 out of 17; 94%) can be considered as ‘independent’ observations, as they come from different rats. We also sought to test whether active neurons were ‘clustered’ in some rats. To this end, we focused on the subset of animals with ≥ 5 recordings (n=15 rats); we find that the occurrence of active neurons was not significantly different from their expected proportion from the population data (~14%; see corresponding p values from binomial test in Figure 5).

Author response image 1.Distribution of recordings across animals.Histogram showing the distribution of active (red) and silent recordings (blue) across animals. In the animals with a relatively larger number of recordings (≥5, n=15 rats), the occurrence of active neurons was not significantly different from their expected proportion from the population data (~14%). Corresponding p values from binomial test are shown.**DOI:**
http://dx.doi.org/10.7554/eLife.20252.012

Altogether this indicates that – considering also that animal age, housing and behavioral training were standardized across animals – the relationship between morphology and activity is unlikely to reflect a bias in a subset of animals. However, we acknowledge that, given the relatively small number of observations per rat, a dependency between rat identity and morphological properties cannot be rigorously tested and thus formally ruled out. We specify this in the revised manuscript (see below).

To address this point, we have included a new Figure 4—figure supplement 2. We refer to this analysis in the revised manuscript (Results, eighth paragraph and Methods, subsection “Targeting of the dentate gyrus and dataset of juxtacellular recordings”, second paragraph). We specify that a relationship between morphology and rat identity cannot be formally ruled out in our dataset (Figure legend of Figure 4—figure supplement 2).

*4) The description of the classifier and its evaluation is not sufficient. What properties of the spike waveform were used for training? What was the exact composition of the training and test datasets? Is it just the 47 neurons in Figure 1? This is a somewhat small number of samples compared to datasets often used for training classifiers. What was the false positive and false negative rate when testing the classifier? What was the quantitative criterion for shoulder or no shoulder in Figure 1? Figure 1 indicates that the shoulder is variable and not present in all granule cells. Does this have implications for interpretation of the data and what are these?*

We apologize for not having provided sufficient information about the classifier. We now clarify that (i) the classifier was trained on average spike waveforms, that (ii) the classifier was trained on our dataset of identified neurons (n=47; as in Figure 1), with a leave-one-out cross-validation, and (iii) we provide the false-positive and false-negative rates (false-positive: 2/47 = ~4%; false-negative: 3/47 = ~6%). We also acknowledge in the revised manuscript that, in line with the reviewers’ comment, our classification results rest on a relatively small training dataset. This is specified in the revised Results.

We would also like to clarify the rationale behind the use of the classifier in our study. Classification of neurons as ‘putative GCs’ is not strictly dependent upon the neural-network classifier approach; recordings are classified as ‘putative’ GCs by the occurrence of spike- shoulders, which were detected by the intersection between the spike trace and its moving average during the repolarization phase of the spike (more details are provided in the new Figure 1—figure supplement 1). The neural-network classifier was implemented for testing whether recordings can also be assigned to the GC class based solely on their average spike waveforms. We believe that in this respect, the classification results provide additional evidence for a relationship between spike-shape and GC identity. This is now specified in the revised Results.

In line with the reviewers’ comments (i.e. ‘What properties of the spike waveform were used for training?’),we sought to determine which specific features of the spike-waveform are used by the neural-network classifier. We found that training the classifier only on the repolarization phase of the average spike waveforms (i.e. where spike-shoulders occur) provides an equally- high classification accuracy (~87%), suggesting that spike-shoulders are an important determinant of the classifier performance. This is stated in the revised Results.

As for the last point, indeed shoulders are variable (i.e. they seem to vary in prominence and location after the spike-peak) and are not present in all GCs (see below). We have performed additional analysis (shown in Figure 1—figure supplement 1) where we quantified the prominence and position of spike-shoulders relative to the spike-peak. These two parameters were not significantly different between identified versus classified GCs (see Figure 1—figure supplement 1). We also state that since most but not all identified active GCs displayed spike-shoulders (18 out of 21; ~86%, Figure 1), a fraction of GC recordings (~14%) is likely to be misclassified as false-negative in our dataset.

We refer to these in the revised manuscript (Results, first and fourth paragraphs; Methods, subsection “Analysis of Electrophysiology Data”, last paragraph) and we added an explanatory additional figure, Figure 1—figure supplement 1.

*5) Figure 4. The legend indicates that "the distributions" are statistically different. Is this referring to length, distance, or length data as a function of distance as shown in the figure? The conclusion is based on results of a Mann Whitney U which would usually be used to test for a difference in medians. It is not clear how this was applied here.*

We agree with the reviewers that the comparison of length data between active and silent neurons was not appropriate in the context of Figure 4. We have now compared dendritic lengths in each bin (10 μm) between active and silent neurons (Mann-Whitney U test). We display the differences at 1% significance level (two stars) in revised Figure 4.

We updated Figure 4 and the corresponding figure legend.

*The analysis of morphology and its presentation needs to be improved. Assuming that the morphologies of the 7 silent and 6 active granule cells analyzed in detail are complete and representative, the main conclusions of the study are fairly straightforward. Therefore, it is important for the authors to devote more of the manuscript to their morphological methods and clarify their procedures in more detail.*

Along these suggestions, we have performed additional analysis (see details below), which altogether indicates that our sample of identified GCs is likely to represent a random sample of active and silent neurons, and that the morphological differences are unlikely to result from methodological/procedural biases. We thank the reviewers for this comment, since it resulted in a more rigorous presentation of our findings and it allowed us to greatly strengthen the conclusions of our work.

*6) The authors write "we reconstructed the somato-dendritic compartment of 7 silent and 6 active GCs, which were selected for the morphological analysis due to their high-quality filling and complete dendritic morphology." What were the criteria for assessing the completeness of the dendritic morphology? Was the labeling protocol standardized in some way between silent and active neurons?*

In the previous version of the manuscript, we had referred to qualitative criteria for assessing the quality of neuronal labeling (e.g. qualitative appearance of staining intensity, dendritic segments reaching the outer molecular layer, etc…). To address this comment rigorously, we have now quantified and compared the labeling/filling quality between active and silent neurons. This was done by 3 approaches: first, we compared the efficiencies of juxtacellular labeling. As specified in the revised Results, the same labeling procedures were applied to silent and active neurons, which resulted in similar efficiencies of dendritic filling (active, 6/14 labeling attempts, ~42%; silent, 7/19 labeling attempts, ~37%; p=1). Second, we quantified labeling/filling quality by measuring the gray levels of proximal and distal dendritic compartments in active and silent neurons. Incomplete filling typically results in weaker staining intensities, which tend to become weaker as a function of distance from the soma (a ‘fading-off’ profile). We found that both gray values at distal dendritic compartments and distal versus proximal intensity ratios were not significantly different between active and silent neurons (Figure 4—figure supplement 2). Moreover, no significant correlation was found between the labeling intensities and dendritic complexity (p = 0.32). Third, we compared the distribution of dendritic endings as a function of distance from the soma. We reasoned that incompletely- filled neurons should display dendritic endings located closer to the soma; this distribution (together with total dendritic lengths, see [Supplementary-material SD2-data]) was not significantly different between active and silent neurons (see Figure 4—figure supplement 2). Altogether this analysis indicates that labeling biases are unlikely to account for the differences in morphological parameters between active and silent neurons.

As for the completeness of dendritic morphologies, we acknowledge the fact that complete filling cannot be determined with certainty. While the new analysis indicates that filling/labeling quality was not significantly different between active and silent cells (Figure 4—figure supplement 2), the completeness of dendritic morphologies is always an assumption which is difficult to prove experimentally even in the best filled examples. This is now stated in the revised Methods.

We included one additional figure (Figure 4—figure supplement 2) and we refer to this point in the revised manuscript (Results, eighth paragraph; Methods, subsections “Histochemistry and cell reconstruction” and “Analysis of Morphological Data”).

*For instance, is it possible that it was harder to fill silent cells than active ones? What is the fraction of silent cells for which labeling was attempted whose dendritic tree was completely filled, and how does that compare to the fraction for active cells? If there was any difference in the rate of complete filling, could it be possible that there was a bias in the labeling method that could lead to an apparent difference in the level of branching or complexity? Is it possible, for example, that silent cells were harder to fill and more likely to have somewhat less complete dendritic fills, and thus more likely to appear to have less complex trees?*

To address this comment, we have compared the labeling efficiencies between active and silent neurons. In our dataset, silent and active cells appeared to be labeled with similar efficiencies; in fact, the proportion of successful labeling attempts (active: 9 identified cells out of 14 labeling attempts, silent: 8 identified cells out of 19 labeling attempts; p = 0.56) as well as the proportion of neurons with high-quality filling (active: 6 out of 9 identified, silent: 7 out of 8 identified; p = 1) were not significantly different between active and silent neurons. We refer to this point in the revised manuscript. These results are in line with the ones referred to in response to the above comment.

We refer to this point in the revised manuscript (Results, eighth paragraph and Methods, subsection “Analysis of Morphological Data”).

*7) The branching and complexity measures that show a difference between active and silent cells are defined in the methods: "The 'branching index' was defined as the number of dendritic endings divided by the number of primary dendrites. The 'complexity index' was computed as in previous work (Pillai et al., 2012) according to the following equation: (sum of branch tip orders + number of branch tips) x (total dendritic length/total number of primary dendrites)." These measures appear to be particularly sensitive to the number of primary dendrites as it appears in the denominator in both cases. Therefore, the authors should include in the table mentioned above, for each of the 13 neurons with complete dendritic trees, the firing rate in the arena, the branching and complexity values, the total dendritic length, the number of dendritic endings, and the number of primary dendrites. The authors write that there was a non-significant 2x difference in the number of primary dendrites "(the number of primary dendrites (active, 1.1+/-0.4; silent, 2.2+/-1.1; p=0.08)", but the branching index also differed by 2x "(branching index; active, 14.6+/-4.6; silent, 7.8+/-4.7; p=0.026)", though this difference was significant. This suggests that the number of dendritic endings is similar for both silent and active cells. Therefore, is it possible that the difference in the number of primary dendrites could be responsible for differences in activity, instead of the difference in spine number that they suggest as the cause? In comparison, the number of spines differed by a factor of 1.4x "(total number of spines; active, 1802+/-353; silent, 1354+/-257; p=0.035)" and they state "The total number of dendritic spines also showed a significant correlation with firing rates (r=0.63; p=0.019; n=13)". What is the correlation between the number of primary dendrites and firing rate and associated significance?*

We thank the reviewers for this comment. Indeed, the number of primary dendrites showed a two-fold difference between active and silent neurons (which did not reach significance in our dataset). We take this as an indication that active and silent GCs might indeed represent fundamentally-different morphologies at the opposite end of the morphological spectrum. Whether this difference per se can translate (and account for) the differences in firing activities cannot be determined in our dataset, as we do not find a significant correlation between the number of primary dendrites and in-vivo activity (r=-0.49; p=0.089; now reported in the revised Results) Additional work will be required to determine whether integrative processes, which more critically depend on the number of primary dendrites and dendritic branch architecture (e.g. Xu et al., 2012; Sheffield and Dombeck, 2015) correlate with somatic spiking.

In the manuscript (Discussion, fourth paragraph) we speculate about a possible causal relationship between in-vivo GC activity and the total number of spines contributed in the input layer. This speculation is based on the significant correlation between firing activity and dendritic complexity (r = 0.68, p = 0.009) as well as the total number of spines (r = 0.63, p = 0.019). In our view, this appears as the most parsimonious interpretation of our data, also in light of previous work, where a relationship between dendritic architecture, spine density and activity has been proposed (e.g. Washington et al., 2000; Krichmar et al., 2002; Oberlaender et al., 2012; Roy et al., 2016). We acknowledge however that our interpretation is at present speculative, and that other cell-autonomous and/or network mechanisms (e.g. intrinsic properties, spatio-temporal dynamics of inhibition) are likely to play an important role in determining GC activity. This is specified in the revised Discussion.

In line with the reviewers’ comment, we included the morphological parameters in the summary table ([Supplementary-material SD2-data]). The above points are referred to in the revised manuscript (Results, fifth and seventh paragraphs; Discussion, fourth paragraph).

*8) The authors write that "The large majority of sampled neurons were silent (see Figure 2). In order to record from spiking GCs, silent recordings were routinely discarded by further advancing the electrode within the layer." Does this mean that their sample was biased toward more active cells? If so, they should clearly point this out when they state proportions of cells that were active. For instance, they state "Active GCs were thus very sparse, consistent with previous estimates (Jung and McNaughton, 1993; Leutgeb et al., 2007; Neunuebel and Knierim, 2012) and accounted for only ~14% (32 out of 228) of all blindly-sampled neurons within the GC layer." If they biased their search for active cells, then they should remove "blindly-sampled" from this sentence and point out that the 14% value is an overestimate based on their sampling method. In addition, is it possible that any bias in the search toward active cells could influence the morphological results? For instance, is it possible that cells that are active during the search and active during exploration of the arena are not a random sample of the entire set of cells active during exploration? Perhaps these cells have more branching and complex dendritic trees compared to cells that are active during exploration but were not spontaneously active during the cell search procedure? The basic question is whether the sample of 7 silent and 6 active cells analyzed in detail morphologically constituted an unbiased sample of silent cells, and an unbiased sample of active cells, respectively. If the possibility of such a bias cannot be ruled out, yet there is no reason to think there is a bias, the authors could just make a short comment about how they assume their methods have isolated a random sample of silent and active cells.*

We thank the reviewers for this comment, as it allowed us to better clarify this issue. We specify that neurons were searched while animals were freely-behaving in the arena. Thus, an active neuron during ‘search’ is an active neuron during ‘exploration. In this respect, our search procedures can be considered ‘blind’; however – in line with the reviewers comment – we acknowledge that the proportion of active neurons is likely to represent an overestimate. This is now stated in the revised manuscript (see below).

Our analysis however indicates that it is unlikely that systematic procedural biases could have accounted for the morphological differences between active and silent neurons. In fact, many parameters were similarly distributed among the two populations (see Results and [Supplementary-material SD2-data], Figure 4—figure supplement 2), indicating that active and silent neurons are likely to represent a random sample of the GC population. We note however that, in line with the reviewers’ comment, we cannot formally exclude biases of the ‘blind’ juxtacellular sampling procedures. Despite (i) being a method largely unbiased to many cellular features (e.g. soma size, activity, cell packing density), and (ii) our previous experience in parahippocampal cortices, where juxtacellularly-labeled neurons were typically recovered at their expected anatomical ratios (Burgalossi et al., 2011; Ray et al., 2014; Preston-Ferrer et al., 2016; Tang et al., 2016), we cannot formally exclude the possibility that biases –intrinsic to the juxtacellular method – do exist. This is now specified in the revised manuscript.

We refer to our search procedures and possible methodological/procedural biases in the revised Results (seventh paragraph) and Methods (subsection “Targeting of the dentate gyrus and dataset of juxtacellular recordings”, second paragraph). In line with the reviewers’ comment, we have also removed the expression ‘blindly-sampled’, and stated that our proportion of active neurons is likely to represent an overestimate (Results, third paragraph; Methods, fifth paragraph).

*How long did they wait before discarding silent cells? This defines an upper bound on the possible spike rate of silent cells.*

Following the reviewers’ comment, we have now implemented a cutoff threshold and included in the freely-moving dataset only recordings which lasted for > 60s. This sets an upper bound estimate on the possible firing rate of silent GCs to 0.016 Hz.

We specify this in the revised Methods (fifth paragraph).

*The authors recently showed that they could induce place fields in some granule cells by juxtacellular electrical stimulation (Diamantaki et al. 2016). Does this effect also correlate with dendritic structure? Please discuss.*

We agree that this is an important point. Unfortunately, in our previous study we could not collect sufficient morphological data for addressing this point, since the study was primarily designed for single-cell stimulation and not for cell identification. We refer to this comment in the revised Discussion (see below).

In the Discussion we state that ‘We have previously reported that in a subset of silent GC, single-cell stimulation can be sufficient for inducing spatial activity (Diamantaki et al., 2016). It would be interesting to resolve whether this form of plasticity – and more generally transitions between the silent and active pool (Lisman, 2011) – are also correlated to dendritic morphology. A combined single-cell stimulation and cell-identification approach will be required for testing this hypothesis’.

*The paper should be more tightly framed in terms of measurement of sparseness (additional literature should be cited) and structure-function-relationship.*

We have shortened our Introduction, re-shaped the flow towards structure-function relationships and referred to the key literature about sparse activity.

Additional references have been cited in the revised Introduction (last paragraph) and Discussion (first paragraph).

[Editors’ note: the author responses to the second round of peer review follow.]

*[…] We note that the reviewers remain very positive about the data set and the finding of sparse coding in the dentate gyrus. However, the focus of the manuscript is the link between activity and dendritic structure (the title is 'Structural correlates of granule cell activity in the dentate gyrus of freely-moving rats'). Given this focus all reviewers and the Reviewing Editor remain concerned about the statistical analysis linking activity and dendritic structure. Some commonly used measures of dendritic complexity, such as dendritic length and the number of primary dendrites, showed no significant effect, whereas some less intuitive measures (i.e. 'dendritic complexity') did, but at modest levels of significance. Most of the reported measures are related and thus not independent. There are some indications that the study may be underpowered to make conclusions about dendritic structure. We are not convinced that the results were corrected for multiple comparisons together in an appropriate manner. For these reasons we cannot publish your manuscript in its present form.*

However, we would be happy to consider the manuscript again with one of the following major revisions:

1) A more compelling statistical analysis. This could take the form of a rigorous analysis of multiple comparisons. Alternatively, a reanalysis with randomly chosen 50% of the data might also be appropriate.

2) Addition of data that make the conclusions about structure-function relationships stronger.

*3) A refocusing on sparse coding in the dentate gyrus.*

We thank the reviewers for their constructive suggestions, and for the opportunity of considering a major revision of our work. Statistical re-analysis (see point 1 below) have involved Dr. Philipp Berens (Neural Data Science group, Bernstein Center for Computational Neuroscience, University of Tübingen), whose contribution has resulted in a co-authorship.

Based on the reviewers’ feedback, we have implemented the following changes:

1) *We have completely revised our statistical analysis* of the relationship between cell type (silent or active) and morphological properties of the cells. Indeed, the reviewers were right that within the classical hypothesis testing framework it makes a big difference whether a study has a preregistered analysis plan for looking for associations between a large number of predictors (in our case morphological features) and a target variable (in our case silent/active) or whether it is an exploratory study, where the specific analysis is determined during the course of analysis. As our study is explorative by nature, this is a potential issue.

To make our statistical analysis more rigorous and determine the relationship between morphological features and cell type (active vs. silent), we departed from the hypothesis testing approach and asked whether silent and active neurons could be discriminated by a classifier based on morphological features. This classification approach is particularly appealing as it provides a very intuitive notion of the effect size and the predictive power of the morphological features (in contrast to the hypothesis testing approach, which provides only a binary decision).

We used nested leave-one-out cross-validation with a logistic regression classifier (as in Jiang et al., 2015; Cadwell et al., 2016) and primary morphological measures as features (see revised Figure 4). With the so-called “elastic-net penalty” (enforcing different levels of sparsity in the weights) this classifier is very well suited to cope with correlated features (Friedman et al., 2010). The cross-validation procedure entails that a classifier is trained on all but one cell and tested on that held-out cell, to avoid reporting an overly optimistic estimate of the classification performance. The classifier was able to discriminate silent and active cells in up to 85-92% of the cases, depending on the level of sparseness enforced (11-12 out of 13 cells classified correctly). To assess whether this could have occurred by chance we ran a permutation test, shuffling the cell type labels and rerunning the classification analysis (500 runs). This analysis indicates that the classification performance achieved by our classifier was very unlikely to have occurred by chance (P=0.008).

Studying the weights used by the classifier allowed us to analyze which morphological features were used by the classifier to make its predictions and are therefore associated with the silent/active distinction. We found that the length of high-order dendrites (5 and 6) was most predictive of active cells, while the length of low-order order dendrites (2 and 3) as well as the number of primary dendrites were most predictive of silent cells (revised Figure 4). Our analysis also addresses the following reviewers’ concern: ‘[…] Some commonly used measures of dendritic complexity, such as dendritic length and the number of primary dendrites, showed no significant effect, whereas some less intuitive measures (i.e. 'dendritic complexity') did’.The classifier performance was in fact determined by primary dendritic measures (Figure 4); the inclusion of derived morphological measures – like e.g. ‘branching index’ and ‘dendritic complexity’ (see [Supplementary-material SD2-data])- did not further improve classification accuracy. This indicates that in our dataset of identified neurons, primary measures of dendritic architecture alone contain sufficient information for classifying neurons into active and silent by logistic regression.

Overall the results of this analysis are in good agreement with the results we previously reported – indicating that active cells have higher dendritic complexity – and thus further strengthen the conclusions of our study. We are convinced that this analysis addresses the issue brought up by the reviewers. The code for running this analysis is available as part of our submission.

2) *We have re-focused on sparse coding in the dentate gyrus*. We now present our work in a more concise format (as a ‘Short Report’), we acknowledge that the study is exploratory and that the morphological findings rest on a limited dataset of identified neurons – in line with the reviewers’ suggestion for a more rigorous interpretation and balanced account of our data.

Reviewer # 1:

*[…] Figure 4 seems to have 11 points, not 13. It is hard to believe that the p value is < 0.01*

We apologize for the poor display of data points in our former Figure 4. Two points from silent cells were not missing, but overlapping. This plot is not shown in the revised manuscript (source data are however provided).

*Reviewer #2:*

*[…] Different multiple comparison issues apply to Figure 4. I also don't understand how total dendritic length can be similar between the two groups, but the area of the plots in 4D looks quite different.*

We apologize for not having clarified this point in the previous version of the manuscript. Our former Figure 4 was in fact related only to high-order dendrites (i.e. orders ≥4; as stated in the corresponding figure legend). In the revision, we show an improved version of the figure.

*Additional comments:*

*The table only includes a subset of the measurements used for the analysis. It would be more helpful if all measurements are included.*

We now include a revised Figure 4—source data, where all measurements of dendritic structures and electrophysiological parameters are included. In addition, we provide the code for running the analysis for Figure 4 on.

*It's not clear why the Abstract reports correlations rather than results of comparisons between active and silent neurons.*

We thank the reviewer for this comment. We have revised the Abstract.

*The recording duration is longer for active than silent cells. Is this an issue?*

We believe the impact of relatively shorter duration of silent recordings is in setting an upper-bound limit on the possible firing rate of silent neurons. This limitation is referred to in the revised manuscript (subsection “Targeting of the dentate gyrus and dataset of juxtacellular recordings”, fourth paragraph).

*Reviewer #3:*

*The authors have adequately addressed my questions and concerns. I have a few remaining comments:*

*From the table, the numbers that stand out most are the differences in the number of primary dendrites, even though this did not reach significance.*

The reviewer is right. Weights analysis of our logistic regression classifier (see revised Figure 4) indicated that the number of primary dendrites has some influence on classifier performance at least when little sparseness is enforced – in line with the Reviewer’s comment (Figure 4). This is now stated in the revised manuscript (Results, last paragraph).

*In the Abstract, the authors write "We found that the majority of neurons (163 of 190) were silent during exploration." However, I recommend that the authors remove the numbers since it reads as if it is an unbiased estimate of the fraction of silent neurons, but they confirmed in the response that this is likely to be underestimate due to their search procedure. Instead they could add "vast" in front of "majority" to make their point.*

This point has been addressed in the revised manuscript.

*In the Figure 2 legend the authors write "Unlike extracellular recordings, juxtacellular sampling is not biased towards active cells, since silent neurons can also be recorded and their presence confirmed by current injection." This is followed by "Cumulative plot showing the firing rate distribution within the GC layer. Each red circle represents one neuron, sampled juxtacellularly within the GC layer (see Methods for details). Note the large proportion of silent neurons (163 out of 190) compared to active cells." However, as acknowledged by the authors, their sampling of the proportion of active and silent cells is likely to be biased due to their search procedure, where inactive cells were often discarded early during exploration, before the 60-second threshold they used for counting the cells. Because of this, I think this statement and the numbers could confuse the readers into thinking that these numbers are an unbiased estimate, and one that is a better estimate than obtained with other methods. It would be a better estimate if their cell search / counting procedure was unbiased, but apparently it was not. The correct proportion is an important number. The authors should therefore clarify this in the legend as they have done in the main text. For example, they could write in the legend "Note the large proportion of silent neurons (163 out of 190) compared to active cells. Furthermore, this proportion is likely to be an underestimate of the true silent proportion due to the details of the search procedure (see text and Methods)."*

This is a fair point. We have addressed it in the revised manuscript as suggested by the reviewer (see Figure 2 legend).

[Editors' note: what now follows is the response after the authors submitted for reconsideration.]

*[…] Essential revisions:*

*Although the classifier can use structure to predict function, the relationship is subtle and can't clearly be boiled down to simple measures (at least given the limited data set) (see also Figure 4—figure supplement 1). It would be good to make this last point clearer (the relevant statement in the Discussion is not accurate and should be revised).*

We made this point clearer in the revised Discussion (last paragraph), where we also state the following: ‘We note however that in our limited dataset of identified GCs (n=13) structure-function relationships are subtle, as they derive from the combined analysis of multiple structural parameters (Figure 4)’.’

*Describe more clearly what a primary, as in 1st order, dendrite compared to a 2nd order dendrite. This is because (1) it appears from the table that this this would have a large effect on distinguishing active and silent cells, not just in terms of the total 1st order length, but also in terms of the total higher order lengths (since a 5th order branch would be a 4th order branch if the 2nd order branch it came from was instead called a 1st order branch), and (2) the table shows that the 1st order branches are generally short. Therefore, in the Figure 4—figure supplement 1 the authors should include beside each neuron a close-up of the somatic region showing the soma, 1st order branch(es), and start of 2nd order branches, ideally with markers to show where the divisions are.*

We agree with the reviewers that this is indeed an important point. To address this rigorously, we have performed additional analysis (see points 1-3 below).

1) We have double-checked and re-traced the primary dendrites of all neurons (n=13), and used an alternative measure for the primary dendritic length (see point 2 below). We have also corrected one mistake in the previous version of the manuscript: cell id 103 was erroneously displayed as having 2 primary dendrites, instead of 1. For clarity, we show this neuron in Figure 6. From the two different focal planes one can clearly see that only one primary dendrite is emerging from the soma. We apologize for the mistake. We note that this change results only in a minor redistribution of the weight profile of the classifier (see revised Figure 4), and the conclusions of our findings remain unaffected (classification accuracy = 85%; see Results, page 6 lines 104ff).

2) We agree with the reviewers that primary dendrites are relatively short (median ~8 µm) – this is however only slightly smaller than values reported in the literature, e.g. Caceres and Stewart, 1983; Zhao et al., 2010). We think this is primarily a result of (i) how primary dendrites are defined and (ii) the fact that our reconstructions were not compensated for tissue shrinkage. In the previous version of the manuscript, ‘primary (or 1_st_ order) dendrites’ were defined as the distance between the soma edge and the first dendritic branching node. GCs with only one primary dendrite often display an elongated soma, which makes it difficult to determine the exact border between the somatic and the primary dendritic compartment. This is schematically shown as Figure 6. Depending on the soma contour, this measure can jitter (from 3.3 to 5.9 µm in the representative example; cell id 103). In the previous version of the manuscript, we had used the shortest measurement across all cells. We note that the same definition and criteria were applied to both active and silent neurons (cells with short primary dendrites are present in both classes; active, cell ids 83, 103, 104, 1046; silent, cell ids 924, 949, 993) and hence unlikely to have biased our results.

Author response image 2.(**A**) Representative pictures of the cell id 103 at two different focal planes.Note the presence of a single apical primary dendrite. The arrow points to the initial segment of a secondary dendrite, which continues in the neighboring section. (**B**) Schematic diagram showing the jitter in estimating the length of short primary dendritic segments, depending on the shape (i.e. more or less elongated) of the somatic contour.**DOI:**
http://dx.doi.org/10.7554/eLife.20252.013

In order to address the reviewers’ comment, we implemented an alternative measure and defined primary dendrites as the distance between the soma center and the first branching node. We found this to be a more consistent measure across experimenters (the longer measurements reduce the relative error) as the soma center can be unequivocally identified as the center of the inner bounding circle. A close-up magnification on the primary dendrites for all neurons is now shown in Figure 4—figure supplement 1. Computing primary dendritic lengths this way leads to virtually identical results as in the previous version of the manuscript, thus the conclusions of our findings remain unaffected (see revised Figure 4).

3) We note that the classifier performance is not critically dependent on the total length of primary dendrites per se, as it can be seen from the weight profile in Figure 4. However, we agree with the reviewers that the definition of 1_st_ order dendrites ‘would have a large effect on distinguishing active and silent cells […] in terms of the total higher order lengths (since a 5th order branch would be a 4th order branch if the 2nd order branch it came from was instead called a 1st order branch)’. This is an important point, which we sought to address with additional analysis. To this end, we simulated an ‘extreme’ scenario, and removed the first branching node from all neurons with short primary dendrites (same cell ids as in point 2 above). This introduced a shift in dendritic orders, e.g. former 5_th_ order became 4_th_ order, and so forth; as pointed out by the Reviewer. Classification accuracy was however largely unaffected, and was only slightly worse for low enforced sparseness (76-85% classification accuracy as a function of enforced sparseness levels). We take this result as an indication that the morphological differences between active and silent cells are largely unaffected by the definition of primary dendrites, but rather reflect a more general difference in dendritic architectures. We note however that – in line with the reviewers’ comment above – these differences are subtle, as they cannot be narrowed down to a single morphological parameter (now stated in the revised Discussion).

It would be valuable to the community for the authors to include some of the data from the original manuscript in the source file on morphological parameters, such as the "soma location within the GC layer" and the "laminar location of the cells (suprapyramidal versus infrapyramidal blade)."

We included these parameters in the revised [Supplementary-material SD2-data].